# Eigenvectors of Experts are Training-free Non-collapsing Routers

**Giang Do** [1]   **Hung Le** [1]   **Truyen Tran** [1]

## Abstract

Sparse Mixture of Experts (SMoE) architectures improve the training efficiency of Large Language Models (LLMs) by routing input tokens to a selected subset of specialized experts. Despite their remarkable success, both training and inference in SMoE models suffer from the *expert collapse* issue (Chi et al., 2022), which degrades model performance. Prior studies primarily focus on improving the router; however, such methods rely on training from scratch or fine-tuning, which requires high computational and data-processing costs. Furthermore, we demonstrate that, despite these efforts, the issue persists when advancing well-pretrained SMoE models, as evidenced by both theoretical and empirical results. To fill that gap, we analyze the advanced SMoE models and observe that the eigenvectors of expert weight matrices encode rich semantic information, pointing to an effective alternative to conventional routing strategies. Building on this insight, we propose **Singular Value Decomposition SMoE (SSMoE)**, a novel and *training-free* framework that leverages spectral properties of the expert weights to address the collapse issue and enhance model performance. Extensive experiments across diverse language and vision tasks, under both clean and corrupt data settings, demonstrate the strong generalization and robustness of SSMoE. Our findings highlight how a deeper understanding of model internals can guide the development of more effective SMoE architectures. Our implementation is publicly available at https://github.com/giangdip2410/SSMoE.

---

[1]Applied Artificial Intelligence Intiative (A2I2), Deakin University, Victoria, Australia. Correspondence to: Giang Do <truong.do@deakin.edu.au>.

*Proceedings of the 43rd International Conference on Machine Learning*, Seoul, South Korea. PMLR 306, 2026. Copyright 2026 by the author(s).

## 1. Introduction

Sparse Mixture of Experts (SMoE) models, which offer strong scalability, have achieved remarkable success across various domains, including Natural Language Processing (NLP) (Fedus et al., 2022; Du et al., 2022; Team, 2024; Muennighoff et al., 2025), visual representation learning (Riquelme et al., 2021; Shen et al., 2023), and multimodal tasks (Li et al., 2025a; Shen et al., 2024). However, understanding the underlying reasons for SMoE's success remains a challenge, which limits its adoption in domains that demand both high accuracy and full explainability, such as medicine, economics, and law. To address this issue, Chen et al. (Chen et al., 2022) provides a theoretical explanation of SMoEs in a simplified two-layer non-linear setting. Meanwhile, XMoE (Chi et al., 2022) highlights the problem of representation collapse in Sparse Mixture models from a theoretical standpoint, and SimSMoE (Do et al., 2025a) further investigates this collapse by estimating it through similarity metrics. Although most researchers regard the router as a crucial component and have proposed numerous enhancements (Dai et al., 2022; Do et al., 2023; Pham et al., 2024), VQMoE (Do et al., 2025b) adopts a different strategy employing discrete representation learning to guide token routing.

> **Research Question**
>
> Is it possible to perform routing in Sparse Mixture of Experts (SMoEs) without a router?

> **Finding 1:** We observe that the eigenvectors of expert weight matrices capture rich semantic information.

We investigate routing behavior in several advanced, well-trained SMoE models (Muennighoff et al., 2025; Dai et al., 2024; Team, 2024) and observe that even pretrained routers continue to exhibit collapse issues—a finding consistent with prior work (Pham et al., 2024). To validate this observation, we evaluate router behavior across ten state-of-the-art MoE-based models ranging from 4B to 120B parameters. Our experimental suite encompasses models of varying scales and architectures, including PhiMoE-Tiny (Li et al., 2025b), OLMoE (Muennighoff et al., 2025), Qwen1.5-MoE (Team, 2024), and DeepSeek-MoE (Dai et al., 2024).

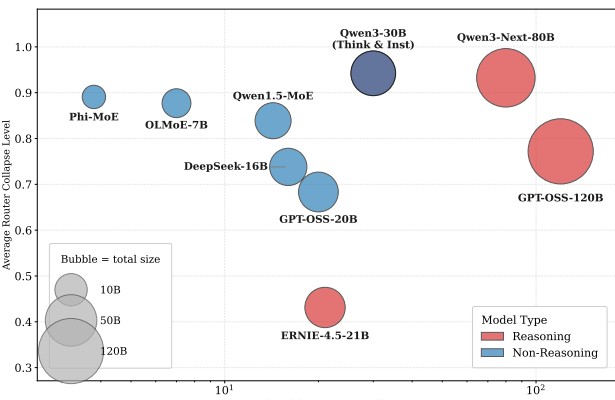

*Figure 1.* Average router collapse levels across layers for ten state-of-the-art MoE-based LLMs. The results demonstrate that while all models exhibit router collapse, the intensity varies between reasoning and non-reasoning models. Best viewed in color.

We also evaluate recent large-scale models such as GPT-OSS-20B, GPT-OSS-120B (OpenAI et al., 2025), ERNIE-4.5 (Baidu-ERNIE-Team, 2025), and the Qwen3-MoE series (Instruct, Think, and Next variants) (Team, 2025). Notably, **all models exhibited router collapse to varying degrees** as Figure 1. This motivates our research question: *Is it possible to perform routing in Sparse Mixture of Experts (SMoEs) without router?* To address this question, we conduct an in-depth analysis of expert weights, typically implemented as two or three Feedforward Neural Network (FFN) layers, through the lens of Singular Value Decomposition (SVD). Our observations reveal that the eigenvectors of these weight matrices encode rich semantic information. Building on this insight, we propose *Singular Value Decomposition SMoE (SSMoE)*, a novel and efficient framework that leverages the eigenvectors of expert weight matrices. These eigenvectors not only serve as effective semantic representations but also contribute to improved token routing strategies. To validate the generalization ability of our approach, we evaluate SSMoE in a *training-free* setting across multiple scenarios—including Large Reasoning Models, Large Language Models, and Vision-Language Models, and with both clean and corrupted data. SSMoE consistently outperforms baseline methods across all three domains, with particularly notable improvements on tasks requiring deep semantic understanding. Notably, SSMoE surpasses the state-of-the-art open-source LLMs - the GPT-OSS models (OpenAI et al., 2025), with an average improvement of approximately 6%, while simultaneously reducing memory consumption by roughly 23% as Figure 2. In summary, this paper makes the following key contributions: (1) We investigate the **expert embedding collapse** issue in advanced SMoE models; (2) We observe that the **eigenvectors** of expert weight matrices encode **rich semantic information**; (3) We introduce **SS-MoE**, a novel and efficient framework that exploits these spectral properties to enhance both expert representation and

token routing; (4) We propose an *expert dropping* method based on SSMoE that improves Sparse Mixture of Experts **performance** while **reducing memory** usage. Our findings underscore the importance of understanding the internal structure of SMoEs, and demonstrate how spectral insights can lead to more effective and efficient architectures.

## 2. Related Work

**Sparse Mixture of Experts (SMoE).** Sparse Mixture of Experts (SMoE), a scalable extension of the original Mixture of Experts framework (Jacobs et al., 1991; Jordan & Jacobs, 1994), has demonstrated strong empirical performance in both Natural Language Processing (Fedus et al., 2022; Du et al., 2022; Team, 2024; Muennighoff et al., 2025) and computer vision (Riquelme et al., 2021; Shen et al., 2023). Despite these advances, SMoE models are frequently affected by representation collapse (Chi et al., 2022), a phenomenon where different experts converge to produce similar or redundant outputs, thereby reducing the effective capacity of the model.

Several approaches have been proposed to address this issue. XMoE introduces low-dimensional routing to promote expert diversity (Chi et al., 2022), while methods such as HyperRouter (Do et al., 2023) and StableMoE (Dai et al., 2022) focus on improving routing stability and reducing variance in expert selection. However, representation collapse remains a persistent challenge, as recent studies continue to report its detrimental impact on model expressiveness and performance (Pham et al., 2024; Do et al., 2025a). In addition, theoretical analyses (Chen et al., 2022) and discrete routing strategies (Do et al., 2025b) have further contributed to understanding the underlying causes of collapse and developing potential mitigation strategies.

**Singular Value Decomposition (SVD).** Singular Value Decomposition (SVD) has proven to be an effective tool for analyzing and fine-tuning Large Language Models (Wang et al., 2025b;a). Prior work by Thamm et al. (Thamm et al., 2022) highlights that critical information in neural network weights is often concentrated in the largest singular values and their corresponding eigenvectors. In contrast, Staats et al. (Staats et al., 2026) demonstrate that although small singular values may seem negligible during pretraining, their removal after fine-tuning can substantially impair model performance. Building on this line of research, Sun et al. (Sun et al., 2025) propose adjusting the scale of singular values in weight matrices to construct expert vectors, each of which specializes in a distinct task type. Recent works such as SMILE (Tang et al., 2026) and MASS (Crisostomi et al., 2026) use eigenvector-based subspace analysis for model merging by decomposing fine-tuning updates $\Delta W$ in task-specific image models. In contrast, SSMoE targets router collapse in a single pretrained MoE by decomposing ex-

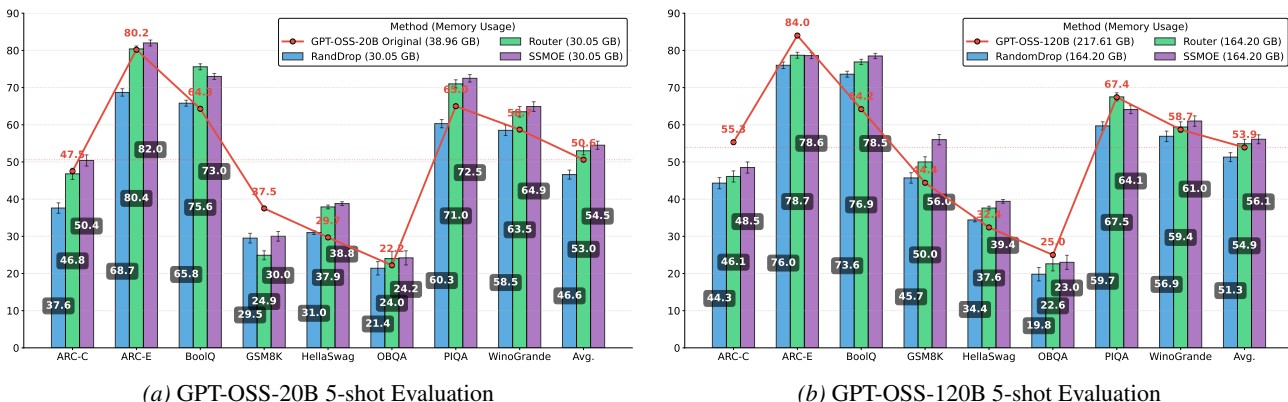

*(a)* GPT-OSS-20B 5-shot Evaluation        *(b)* GPT-OSS-120B 5-shot Evaluation

*Figure 2.* We report performance comparisons across benchmarks for different GPT-OSS model scales under the 5-shot evaluation setting. The proposed SSMoE consistently outperforms all baseline methods across the eight datasets, achieving an average improvement of approximately 13%. Notably, SSMoE surpasses the original GPT-OSS models with an average gain of about 6%, while simultaneously reducing memory consumption by roughly 23%.

pert weights $W$ to capture intrinsic expert specialization in general-purpose LLMs and VLMs.

To the best of our knowledge, this is the first study to investigate the eigenvectors of expert weight matrices in SMoE models and propose their use as an alternative to conventional routers. Our approach not only offers robust and semantically rich representations but also enhances expert selection effectiveness.

## 3. Methodology

### 3.1. Preliminaries

**Sparse Mixture of Experts.** The Sparse Mixture of Experts (SMoE) architecture is a scalable extension of the standard Transformer model that introduces conditional computation by replacing dense multi-layer perceptrons (MLPs) with a collection of specialized subnetworks known as experts (Shazeer et al., 2017). Rather than activating all experts simultaneously, SMoE employs a sparse gating mechanism that selects a small subset of the most relevant experts—typically the top-$k$—based on the input. Given an input $\boldsymbol{x} \in \mathbb{R}^{b \times d}$, where $b$ is the batch size and $d$ is the input dimensionality, the SMoE layer computes its output as a weighted combination of the outputs from the top-$k$ experts:

$$f_{\text{SMoE}}(\boldsymbol{x}) = \sum_{i \in \text{TopK}(\mathcal{S}(\boldsymbol{x}), k)} \mathcal{S}_i(\boldsymbol{x}) \cdot E_i(\boldsymbol{x}), \quad (1)$$

Where each expert $i$ is defined by:

$$E^{(i)}(\boldsymbol{x}) = \text{FFN}_1^{(i)} \cdot \sigma\left(\text{FFN}_2^{(i)}(\boldsymbol{x})\right), \quad (2)$$

where $\text{FFN}_1^{(i)} \in \mathbb{R}^{d \times h}$ and $\text{FFN}_2^{(i)} \in \mathbb{R}^{h \times d}$ and $\mathcal{S}(\boldsymbol{x}) = \text{softmax}(\boldsymbol{x} W_e^\top)$ denotes the gating function, which produces a probability distribution over the $N$ available experts

using a learned projection matrix $W_e \in \mathbb{R}^{N \times d}$. Each expert $E_i$ is an independent feedforward neural network, and only the $k$ experts with the highest gating scores $\mathcal{S}_i(\boldsymbol{x})$ are activated for a given input. This sparsity reduces computational cost and memory footprint while maintaining or improving model capacity and performance.

### 3.2. Singular Value Decomposition SMoE (SSMoE)

**Eigenvectors Representation.** We illustrate the concept of *eigenvector representations* in Figure 3. Unlike SMoE, which relies on learnable embeddings for expert selection, our approach selects experts based on the eigenvectors of their weight matrices, which capture rich semantic information. The computation of *eigenvector representations* involves the following four steps:

*Step 1: Eigenvector Extraction.* For each expert $i$, compute:

$$\begin{aligned} A^{(i)} &= \text{FFN}_1^{(i)} \text{FFN}_1^{(i)\top} \in \mathbb{R}^{d \times d}, \\ B^{(i)} &= \text{FFN}_2^{(i)\top} \text{FFN}_2^{(i)} \in \mathbb{R}^{d \times d}. \end{aligned} \quad (3)$$

Let $\{\boldsymbol{v}_j^{(i,1)}\}$ and $\{\boldsymbol{v}_j^{(i,2)}\}$ denote the eigenvectors of $A^{(i)}$ and $B^{(i)}$, respectively.

*Step 2: Top-c Eigenvector Selection.* Given the router embedding $\boldsymbol{r}^{(i)} \in \mathbb{R}^d$, select the top-$c$ most similar eigenvectors from both sets:

$$\begin{aligned} \mathcal{C}_1^{(i)} &= \text{TopC}(\{\boldsymbol{v}_j^{(i,1)}\}, \boldsymbol{r}^{(i)}, c), \\ \mathcal{C}_2^{(i)} &= \text{TopC}(\{\boldsymbol{v}_j^{(i,2)}\}, \boldsymbol{r}^{(i)}, c). \end{aligned} \quad (4)$$

Then compute the average to obtain the spectral embedding for expert $i$:

$$\mathcal{C}^{(i)} = \frac{1}{2}\left(\text{mean}(\mathcal{C}_1^{(i)}) + \text{mean}(\mathcal{C}_2^{(i)})\right) \in \mathbb{R}^d$$

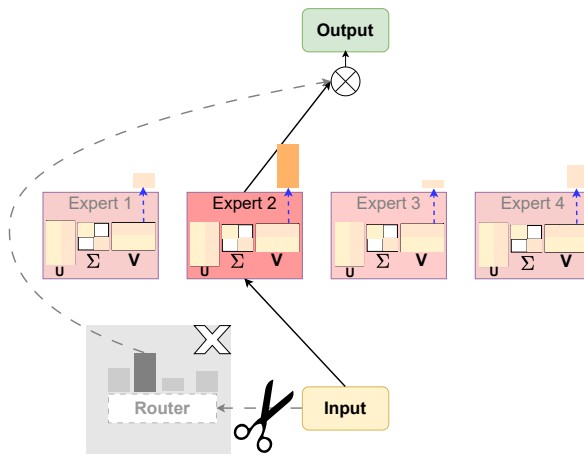

*Figure 3.* An illustration of our *Eigenvectors Representation*, which leverages enriched information from the eigenvectors of expert weights. In contrast, SMoE employs a learnable expert embedding (router) to select the top-$k$ experts for each token. SSMoE provides an efficient and robust representation, as demonstrated in Section 4. Best viewed in color.

*Step 3: Spectral Routing Matrix.* Concatenate the expert spectral embeddings to form the routing weight matrix:

$$W_{\text{EV}} = \begin{bmatrix} \mathcal{C}^{(1)} \, \mathcal{C}^{(2)} \, \dots \, \mathcal{C}^{(N)} \end{bmatrix} \in \mathbb{R}^{d \times N}$$

*Step 4: Eigenvectors Representation.*

$$f_{\text{EV}}(\boldsymbol{x}) = \boldsymbol{x} W_{\text{EV}} \in \mathbb{R}^{b \times N}$$

**SSMoE.** We investigate the eigenvector representations of expert weights, which encode rich semantic information. Prior work (Li & Zhou, 2025) has shown that SMoE routers capture contextually rich features. Building on these insights, we propose a novel architecture that combines both semantic and contextual information by linearly integrating the eigenvector representations with the SMoE router outputs. The detailed algorithm is provided in Algorithm 1. The output of the SSMoE layer for an input $\boldsymbol{x}$ is defined as:

$$f_{\text{SSMoE}}(\boldsymbol{x}) = \sum_{i \in \text{TopK}(\mathcal{V}(\boldsymbol{x}), k)} \mathcal{V}_i(\boldsymbol{x}) \cdot E_i(\boldsymbol{x}), \quad (5)$$

where $\mathcal{V}(\boldsymbol{x}) = \text{softmax}(f_{\text{EV}}(\boldsymbol{x})) \cdot \alpha + \mathcal{S}(\boldsymbol{x}) \cdot (1 - \alpha)$, and $\alpha \in [0, 1]$ is a balancing factor that interpolates between the EV Router and the standard SMoE Router. The hyperparameter $\alpha$ is tuned based on the downstream task data, indicating whether token inputs should rely more on semantic information or contextual information. We empirically observe that $\alpha$ values between $0.5$ and $0.9$ tend to produce optimal results.

## 3.3. Theoretical Analysis

**Lemma 3.1** (SSMoE router reduces pairwise logit correlations: router-independent case). *Let* $f_{\text{EV}}(x) = (f_{\text{EV},1}(x), \dots, f_{\text{EV,N}}(x))$ *be the pre-softmax logits produced by the eigenvector (EV) router and let* $s(x) = (s_1(x), \dots, s_N(x))$ *be the learned SMoE router logits for input* $x$. *Form the convex combination*

$$\mathcal{V}(x) = \alpha f_{\text{EV}}(x) + (1 - \alpha)s(x), \quad \alpha \in [0, 1].$$

*Assume the following conditions:*

1. *(EV orthogonality) For every pair* $i \neq j$, $\mathbb{E}\big[f_{\text{EV},i}(x) \, f_{\text{EV},j}(x)\big] = 0$.

2. *(EV–learned independence) For every* $i, j$, *the EV logits are uncorrelated with the learned logits:* $\mathbb{E}\big[f_{\text{EV},i}(x) \, s_j(x)\big] = 0$.

*Then for every* $i \neq j$ *the pairwise covariance of the combined logits satisfies*

$$\mathbb{E}\big[\mathcal{V}_i(x) \, \mathcal{V}_j(x)\big] = (1 - \alpha)^2 \, \mathbb{E}\big[s_i(x) \, s_j(x)\big].$$

*In particular, when the learned router is collapsed so that* $|\mathbb{E}[s_i s_j]|$ *is large, any choice* $\alpha \in (0, 1]$ *strictly reduces the magnitude of pairwise logit covariance relative to using* $s$ *alone:*

$$\big|\mathbb{E}[\mathcal{V}_i \mathcal{V}_j]\big| = (1 - \alpha)^2 \, \big|\mathbb{E}[s_i s_j]\big| < \big|\mathbb{E}[s_i s_j]\big|.$$

*Thus SSMoE router mitigates collapse (pairwise correlation) of the learned router; the larger* $\alpha$ *the greater the reduction.*

Lemma 3.1 demonstrates that SSMoE alleviates the representation collapse problem commonly observed in conventional SMoE models (Chi et al., 2022). Lemma 3.1 characterizes the router-independent case of SSMoE, including the simple-average EV variant used in our ablation. In this setting, EV descriptors are constructed from expert weights rather than from the learned router decisions, making the decorrelation assumption a plausible analytical model.

**Extension to selective SSMoE.** Our main SSMoE variant further uses the learned router to select a subset of eigenvector directions. This router-guided selection improves empirical performance by filtering misaligned EV directions, but it also introduces mild dependence between $f_{\text{EV}}(x)$ and $s(x)$. Therefore, for selective SSMoE, the exact decorrelation condition in Lemma 3.1 should be interpreted as an approximate decorrelation condition:

$$\big|\mathbb{E}\big[f_{\text{EV},i}(x)s_j(x)\big]\big| \leq \varepsilon, \quad \forall i, j,$$

for a small $\varepsilon \geq 0$. Under this relaxed condition, for every $i \neq j$,

$$\big|\mathbb{E}\big[\mathcal{V}_i(x)\mathcal{V}_j(x)\big]\big| \leq (1-\alpha)^2 \big|\mathbb{E}\big[s_i(x) s_j(x)\big]\big| + 2\alpha(1-\alpha)\varepsilon.$$

Thus, selective SSMoE preserves the decorrelation effect of Lemma 3.1 up to a small cross-correlation error term. This distinction explains why the router-free average EV variant already improves over the original model, while the router-guided variant further improves performance by selecting more relevant eigenvector directions. Empirically, we also observe low overlap between SSMoE and the original SMoE router, supporting that the EV signal is complementary rather than a trivial copy of the learned router. The formal proof is presented in the Appendix B.2.

# 4. Experiments

## 4.1. Large Reasoning Models

**Settings.** We evaluate our method using state-of-the-art open-source MoE LLMs from the GPT-OSS family (OpenAI et al., 2025), specifically two variants: GPT-OSS-20B with 24 layers and 32 experts, and GPT-OSS-120B with 36 layers and 128 experts. Both models activate 4 experts per token. Due to resource constraints, we adopt an expert pruning strategy inspired by expert dropping (Zhou et al., 2025). We propose dropping 25% of experts and compare three dropping strategies: (1) random expert dropping as a baseline; (2) dropping based on average similarity scores between tokens and the router; and (3) dropping based on SSMoE (our method). This approach reduces memory consumption by approximately 23% compared to the original models.

Moreover, we test SSMoE performance across eight established reasoning benchmarks. The ARC-Challenge and ARC-Easy benchmarks (Clark et al., 2018) assess scientific question answering at varying difficulty levels. BoolQ (Clark et al., 2019) evaluates binary reading comprehension through yes/no questions. GSM8K (Cobbe et al., 2021) provides a high-quality dataset of linguistically diverse grade school math word problems. HellaSwag (Zellers et al., 2019) tests commonsense natural language inference (NLI), while OpenBookQA (OBQA) (Mihaylov et al., 2018) probes deeper understanding through question answering that requires reasoning over provided background facts. PIQA (Bisk et al., 2020) measures physical commonsense reasoning by selecting the most plausible outcomes in everyday scenarios, and WinoGrande (Sakaguchi et al., 2021) targets coreference resolution through challenging pronoun disambiguation tasks.

**Reasoning Evaluation.** As shown in Figure 2, SSMoE outperforms the Random Drop approach across all datasets in the 5-shot evaluation setting, achieving an average improvement of **17%** for GPT-OSS-20B and **10%** for GPT-OSS-120B. Notably, compared to the original models, SSMoE surpasses the baseline GPT-OSS by an average of **8%** for the 20B version and **4%** for the 120B version, while utilizing only **75%** of the experts (24 out of 32 experts for GPT-

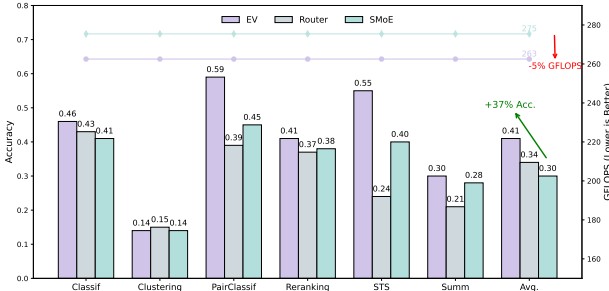

*Figure 4.* We compare the performance of the Eigenvector representation (EV) with traditional routers and *OLMoE-7B* (Muennighoff et al., 2025) on the Massive Text Embedding Benchmark (MTEB). The results show that EV outperforms OLMoE-7B on 37% of the tasks across six benchmarks while reducing computational cost by 5% in GFLOPs. These findings suggest that EV captures rich semantic information. Best viewed in color.

OSS-20B and 96 out of 128 experts for GPT-OSS-120B). This leads to Finding 2: *Not all experts are necessary for reasoning tasks*.

Following the 5-shot evaluation, we test SSMoE under the 10-shot setting without any additional training and report the results in Table 1. The results show that SSMoE outperforms the baseline by an average of **21%** across the eight datasets for GPT-OSS-20B and **10%** for GPT-OSS-120B. Compared to the original GPT-OSS models, SSMoE surpasses the 20B version by **11%** and the 120B version by **4%**, while reducing memory usage by **23%** and **25%**, respectively. Notably, SSMoE outperforms the original GPT-OSS-120B model by **24%** on the BoolQ task and **17%** on the GSM8K task. These results demonstrate that SSMoE excels not only in accuracy but also in memory efficiency.

To disentangle the effect of routing from expert pruning, we also evaluate SSMoE-FULL in the 10-shot setting, where all experts are retained. SSMoE-FULL improves over the original model by **6.4%** on average in Table 1, confirming that the performance gains primarily arise from the proposed SSMoE routing mechanism rather than from reducing the number of experts. Moreover, SSMoE-FULL only slightly outperforms the 75%-expert SSMoE variant by **1.1%**, indicating that expert dropping is not the main source of the improvement. In contrast, naive expert removal through RandomDrop substantially degrades performance. Together, these results demonstrate that SSMoE improves expert utilization through better routing, rather than relying on expert pruning alone.

> **Finding 2:** Certain experts can be omitted from reasoning tasks without compromising performance.

| Model | Method | Memory (GB) | ARC-C | ARC-E | BoolQ | GSM8K | HellaSwag | OBQA | PIQA | WinoGrande | Avg. |
|---|---|---|---|---|---|---|---|---|---|---|---|
| GPT-OSS-20B 10-shot | Original | 38.96 | $44.2_{\pm1.5}$ | $77.9_{\pm0.9}$ | $62.8_{\pm0.8}$ | $\mathbf{36.8}_{\pm1.3}$ | $29.0_{\pm0.5}$ | $19.8_{\pm1.8}$ | $62.5_{\pm1.1}$ | $57.5_{\pm1.4}$ | $48.8_{\pm1.2}$ |
| | SSMoE-FULL (Ours) | 38.96 | $\mathbf{50.2}_{\pm1.5}$ | $\mathbf{79.3}_{\pm0.8}$ | $71.7_{\pm0.8}$ | $43.4_{\pm1.4}$ | $\mathbf{40.9}_{\pm0.5}$ | $\mathbf{33.8}_{\pm2.1}$ | $63.9_{\pm1.1}$ | $58.3_{\pm1.4}$ | $\mathbf{55.2}_{\pm1.1}$ |
| | RandDrop (Baseline) | 30.05 | $33.5_{\pm1.4}$ | $65.8_{\pm1.0}$ | $66.5_{\pm0.8}$ | $33.1_{\pm1.3}$ | $29.7_{\pm0.5}$ | $20.6_{\pm1.8}$ | $57.2_{\pm1.2}$ | $53.3_{\pm1.4}$ | $44.9_{\pm1.2}$ |
| | Router | 30.05 | $48.6_{\pm1.5}$ | $80.1_{\pm0.8}$ | $\mathbf{76.6}_{\pm0.7}$ | $22.3_{\pm1.1}$ | $35.6_{\pm0.5}$ | $24.2_{\pm1.9}$ | $71.3_{\pm1.1}$ | $60.3_{\pm1.4}$ | $52.4_{\pm1.1}$ |
| | SSMoE (Ours) | 30.05 | $\mathbf{52.1}_{\pm1.5}$ | $\mathbf{82.4}_{\pm0.8}$ | $72.4_{\pm0.8}$ | $30.8_{\pm1.3}$ | $\mathbf{37.5}_{\pm0.5}$ | $\mathbf{23.0}_{\pm1.9}$ | $\mathbf{72.2}_{\pm1.0}$ | $\mathbf{62.2}_{\pm1.4}$ | $\mathbf{54.1}_{\pm1.1}$ |
| | vs. Baseline | 0.0% | +55.5% | +25.2% | +8.9% | -6.9% | +26.3% | +11.7% | +26.2% | +16.7% | +20.5% |
| | vs. Original | -22.9% | +17.9% | +5.8% | +15.3% | -16.3% | +29.3% | +16.2% | +15.5% | +8.2% | +10.9% |
| GPT-OSS-120B 10-shot | Original | 217.61 | $\mathbf{52.4}_{\pm1.5}$ | $\mathbf{83.0}_{\pm0.8}$ | $64.0_{\pm0.8}$ | $49.3_{\pm1.4}$ | $30.5_{\pm0.5}$ | $22.0_{\pm1.9}$ | $62.9_{\pm1.1}$ | $56.7_{\pm1.4}$ | $52.6_{\pm1.2}$ |
| | RandDrop (Baseline) | 164.20 | $43.3_{\pm1.4}$ | $74.1_{\pm0.9}$ | $74.2_{\pm0.8}$ | $46.7_{\pm1.4}$ | $33.2_{\pm0.5}$ | $\mathbf{21.4}_{\pm1.8}$ | $56.0_{\pm1.2}$ | $51.8_{\pm1.4}$ | $50.1_{\pm1.2}$ |
| | Router | 164.20 | $42.4_{\pm1.4}$ | $76.1_{\pm0.9}$ | $77.6_{\pm0.7}$ | $49.2_{\pm1.4}$ | $36.2_{\pm0.5}$ | $20.6_{\pm1.8}$ | $\mathbf{65.1}_{\pm1.1}$ | $\mathbf{58.2}_{\pm1.4}$ | $53.2_{\pm1.2}$ |
| | SSMoE (Ours) | 164.20 | $45.6_{\pm1.5}$ | $79.1_{\pm0.8}$ | $\mathbf{79.3}_{\pm0.7}$ | $\mathbf{57.6}_{\pm1.4}$ | $\mathbf{37.1}_{\pm0.5}$ | $21.2_{\pm1.8}$ | $63.3_{\pm1.1}$ | $56.4_{\pm1.4}$ | $\mathbf{54.9}_{\pm1.2}$ |
| | vs. Baseline | 0.0% | +5.3% | +6.8% | +6.9% | +23.3% | +11.7% | -0.9% | +13.0% | +8.9% | +9.6% |
| | vs. Original | -24.5% | -13.0% | -4.7% | +23.9% | +16.8% | -3.6% | +0.6% | -0.5% | +4.4% | +4.4% |

*Table 1.* Performance comparison of different methods on GPT-OSS models with 10-shot evaluation. Results show mean $\pm$ standard deviation across eight benchmarks. SSMoE-FULL denotes our SSMoE variant using 100% of experts, enabling a memory-matched comparison with the original model. Best results in each model size are shown in **bold**. Improvements are calculated relative to baseline methods.

## 4.2. Large Language Models

**Settings.** We evaluate the efficiency and robustness of our method compared to baseline approaches under two scenarios: (1) clean and (2) corrupt.

In the clean setting, inspired by (Li & Zhou, 2025), we apply our method as a plug-in module to three state-of-the-art SMoE models (OLMoE-1B/7B (Muennighoff et al., 2025), Qwen-MoE-7B (Team, 2024), and DeepSeekMoE-16B (Dai et al., 2024)) without any fine-tuning. These models vary in size (from 7B to 16B parameters) and architecture (ranging from 16 to 28 layers and 60 to 64 experts). We assess their performance on a subset of tasks from the Massive Text Embedding Benchmark (MTEB) (Muennighoff et al., 2023), which covers a wide range of downstream sentence embedding applications, including classification, clustering, pair classification, re-ranking, retrieval, semantic textual similarity (STS), and summarization. Following the MTEB evaluation protocol, we use Accuracy for classification, V-Measure for clustering, Average Precision for pair classification, Mean Average Precision (mAP) for re-ranking, Normalized Discounted Cumulative Gain (nDCG) for retrieval, and Spearman's correlation for STS and summarization.

In the corrupt setting, following the protocol in (Nielsen et al., 2025), we evaluate model robustness under adversarial perturbations. Specifically, we inject random "AAA" token sequences into the input to simulate noisy or corrupted data (denoted as Corrupt). Further implementation details and extended results are provided in the Appendix section.

We compare our method against an approach that uses the similarity scores between the router and expert embeddings as hidden representations, referred to as Router Embedding (or simply Router). Additionally, we evaluate against MoEE (Li & Zhou, 2025), which combines Router Embedding with the hidden states of the SMoE model to form its representations. We also include comparisons with prompt-based methods, such as PromptEOL (Jiang et al., 2024), to assess in-context learning capabilities.

**Semantic Evaluation.** We evaluate the semantic quality of our method by extracting embeddings and testing them on MTEB tasks, as shown in Table 2. Overall, SSMoE (ours) consistently outperforms baseline methods across advanced SMoE models, OLMoE-7B, and DeepSeekMoE-16B, achieving performance improvements of **25% to 30%**. SSMoE demonstrates especially strong gains in tasks such as Classification, Clustering, and STS, highlighting its ability to extract high-quality embeddings without the need for fine-tuning or prompts. Interestingly, the Eigenvector-based representation captures rich semantic information, making the method both simple and highly effective—particularly for OLMoE, where it achieves a top score of 40.9%.

**Robust Evaluation.** To evaluate the robustness of our method, we conduct experiments under the corrupt setting, as shown on the right side of Table 2. We observe a consistent trend: complex representations such as those used in SMoE and MoEE experience a significant performance drop when transitioning from clean to corrupt settings. In contrast, lighter representations, including Router and Eigenvector-based (EV) embeddings, demonstrate greater resilience under corrupt conditions. This observation aligns with the "No Free Lunch" theorem (Wolpert & Macready, 1997), which suggests trade-offs between model complexity and generalization across varying data distributions. Notably, our proposed SSMoE achieves state-of-the-art performance on 15 out of 18 tasks in the clean setting and 16 out of 18 tasks in the corrupt setting, highlighting both its efficiency and robustness.

> **Finding 3:** Eigenvectors of expert weights provide an efficient and robust representation.

| Model | Task | Clean | | | | | Corrupt | | | | |
|---|---|---|---|---|---|---|---|---|---|---|---|
| | | Router | SMoE | MoEE | EV | SSMoE | Router | SMoE | MoEE | EV | SSMoE |
| OLMoE-7B | Classification | 41.2 | 43.4 | 41.8 | 45.9 | **50.6** | 36.0 | 36.9 | 36.1 | 36.3 | **39.7** |
| | Clustering | 13.7 | 14.7 | 14.5 | 14.2 | **16.4** | 6.7 | 6.8 | 6.8 | 6.9 | **7.1** |
| | PairClassification | 45.3 | 39.1 | 45.7 | **59.2** | 51.2 | 28.1 | 26.7 | 27.6 | 29.1 | **30.7** |
| | Reranking | 37.5 | 37.4 | 39.5 | 41.2 | **42.4** | 27.8 | 26.8 | 27.7 | 28.2 | **29.3** |
| | STS | 39.9 | 24.1 | 39.9 | **55.3** | 47.0 | 5.1 | 1.9 | 3.6 | 7.5 | **13.4** |
| | Summarization | 28.4 | 20.9 | 29.8 | 29.6 | **30.6** | 24.2 | 24.5 | 25.3 | 26.4 | **28.5** |
| | **Avg.** | 34.3 | 29.9 | 35.2 | **40.9** | 39.7 | 21.3 | 20.6 | 21.2 | 22.4 | **24.8** |
| Qwen-MoE-7B | Classification | 43.8 | 50.3 | 47.7 | 43.4 | **51.6** | 34.4 | 38.3 | 37.0 | 36.9 | **39.8** |
| | Clustering | 13.6 | 27.4 | 25.2 | 13.7 | **31.1** | 5.8 | 8.2 | 7.2 | 6.1 | **8.6** |
| | PairClassification | 45.9 | 46.9 | 51.5 | 47.0 | **55.4** | 26.6 | 27.4 | 30.1 | **31.7** | 30.7 |
| | Reranking | 39.6 | 45.3 | 48.5 | 40.3 | **49.1** | 26.2 | 28.9 | 29.4 | 26.7 | **29.8** |
| | STS | 38.8 | 38.0 | 51.8 | 41.6 | **57.9** | 0.7 | 0.7 | 2.0 | 1.7 | **6.8** |
| | Summarization | 28.3 | 13.4 | **31.2** | 30.3 | 29.6 | 26.7 | 23.0 | 25.5 | 27.9 | **28.5** |
| | **Avg.** | 35.0 | 36.9 | 42.6 | 36.0 | **45.8** | 20.1 | 21.1 | 21.9 | 21.8 | **24.0** |
| DeepSeekMoE-16B | Classification | 43.4 | 46.6 | 44.4 | 44.0 | **48.6** | 36.2 | 36.7 | 36.0 | **40.0** | 39.7 |
| | Clustering | 13.4 | 18.1 | 17.8 | 14.3 | **21.6** | 6.6 | 6.8 | 6.8 | 7.2 | **8.3** |
| | PairClassification | 45.5 | 40.9 | 46.1 | 46.7 | **51.2** | 26.9 | 25.9 | 27.3 | 28.0 | **31.5** |
| | Reranking | 38.5 | 38.9 | 42.2 | 39.2 | **44.9** | 38.8 | 38.3 | 39.0 | 39.5 | **41.0** |
| | STS | 37.7 | 26.3 | 40.2 | 40.0 | **50.1** | 5.1 | 3.7 | 5.7 | 9.1 | **14.3** |
| | Summarization | 24.9 | 22.0 | 24.4 | 28.9 | **29.9** | 27.8 | 25.2 | 28.3 | 28.3 | **28.4** |
| | **Avg.** | 33.9 | 32.1 | 35.9 | 35.5 | **41.1** | 23.6 | 22.8 | 23.8 | 25.4 | **27.2** |

*Table 2.* Performance comparison across MTEB tasks under **clean** and **corrupt** settings. Results are shown side-by-side. The best score in each row (clean or corrupt) is highlighted in **bold**.

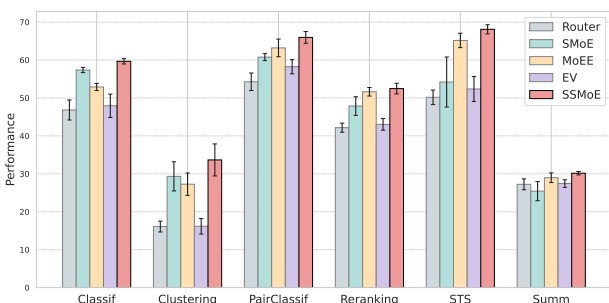

*Figure 5.* Average performance on MTEB tasks across three advanced LLMs (OLMoE-7B, Qwen-MoE-7B, and DeepSeekMoE-16B), comparing SSMoE, SMoE, Eigenvectors (EV), and MoEE using PromptEOL (Jiang et al., 2023) for in-context learning evaluation. Best viewed in color.

**In-context Learning Evaluation.** Figure 5 presents the performance of various methods on MTEB tasks using PromptEOL (Jiang et al., 2024) for in-context learning evaluation. We compare the average performance and variance of SSMoE against baseline methods across three advanced LLMs (OLMoE-7B, Qwen-MoE-7B, and DeepSeekMoE-16B). The results show that SSMoE consistently achieves the highest average scores and the lowest variance among all methods. These findings indicate that SSMoE is not only effective at capturing semantic representations but also excels in in-context learning, underscoring its potential to enhance SMoE-based LLMs on complex tasks such as reasoning.

## 4.3. Vision Language Models

**Settings.** To demonstrate the generalization capability of our proposed method, we evaluate the performance of SSMoE on two tasks: zero-shot image-text retrieval and zero-shot image classification. These tasks assess the model's ability to capture fine-grained visual and semantic information, in comparison to the baseline CLIP-MoE model. CLIP-MoE (Zhang et al., 2025) is a variant of CLIP (Radford et al., 2021) that incorporates a Sparse Mixture of Experts (SMoE) architecture, trained on the ShareGPT4V (ShareGPT) dataset (Chen et al., 2024) to enhance multimodal understanding. For image-text retrieval, we conduct evaluations on the COCO (Lin et al., 2014) and Flickr30k (Plummer et al., 2015) datasets. Following the evaluation protocols for the LLMs section, we test both the efficiency and robustness of SSMoE under two settings: (1) clean images and (2) corrupted images. In the corrupted setting, we introduce random noise to input images, following the procedure described in (Ilyas et al., 2019).

**Zero-Shot Image-Text Retrieval Evaluation.** Table 3 reports results for Image-to-Text (I2T) and Text-to-Image (T2I) retrieval at Recall@1, Recall@5, and Recall@10. Under the clean setting, SSMoE consistently outperforms both the original CLIP model and the CLIP-MoE baseline across all retrieval tasks. Specifically, SSMoE achieves improvements of up to **+1.1%** at Recall@1 on COCO I2T, and up to **+1.0%** on Flickr I2T, demonstrating its enhanced ability

| Setting | Model | COCO I2T | | | COCO T2I | | | Flickr I2T | | | Flickr T2I | | |
|---|---|---|---|---|---|---|---|---|---|---|---|---|---|
| | | @1 | @5 | @10 | @1 | @5 | @10 | @1 | @5 | @10 | @1 | @5 | @10 |
| | CLIP (OpenAI) | 56.1 | 79.5 | 86.8 | 35.4 | 60.1 | 70.2 | 48.5 | 72.6 | 80.8 | 28.0 | 49.3 | 58.7 |
| Clean | CLIP-MoE | 65.0 | 86.0 | 92.0 | 46.8 | 71.7 | 80.4 | 60.5 | 82.3 | 88.8 | 42.1 | 64.7 | 73.2 |
| | SSMoE (Ours) | **65.7** | **86.5** | **92.3** | **47.1** | **72.0** | **80.7** | **61.1** | **82.6** | **89.0** | **42.5** | **65.2** | **73.6** |
| | % Improv. | +1.1% | +0.6% | +0.3% | +0.6% | +0.4% | +0.4% | +1.0% | +0.4% | +0.2% | +1.0% | +0.8% | +0.5% |
| Corrupt | CLIP-MoE | 36.2 | 60.2 | 69.9 | 29.4 | 51.7 | 61.7 | 31.2 | 51.8 | 60.5 | 23.3 | 41.4 | 49.6 |
| | SSMoE (Ours) | **37.2** | **61.0** | **70.3** | **30.0** | **52.1** | **62.2** | **31.6** | **52.1** | **60.7** | **23.6** | **41.9** | **50.1** |
| | % Improv. | +2.8% | +1.3% | +0.6% | +2.0% | +0.8% | +0.8% | +1.3% | +0.6% | +0.3% | +1.3% | +1.2% | +1.0% |

*Table 3.* Performance comparison across different models and settings on COCO and Flickr benchmarks for Image-to-Text (I2T) and Text-to-Image (T2I) retrieval tasks. Best results per column are in **bold**. Percentage improvements of SSMoE over CLIP-MoE baseline are shown with blue color indicating gains.

to capture fine-grained cross-modal relationships.

In the corrupted setting—where random noise is added to the input images—SSMoE maintains its superiority, showing improved robustness compared to CLIP-MoE. Notably, SSMoE yields gains of **+2.8%** at Recall@1 for COCO I2T and consistent improvements across all other metrics and datasets. These findings confirm the effectiveness of the SS-MoE design in improving both accuracy and robustness for zero-shot retrieval tasks, without requiring any additional training.

**Statistical Significance Test.** We observe that SSMoE consistently improves over the original CLIP-MoE baseline across datasets, retrieval metrics, and retrieval directions. While the absolute gains on clean benchmarks are modest, ranging from approximately 0.2 to 1.1 points, they are meaningful because SSMoE is training-free and introduces no additional optimization cost. To verify that these improvements are statistically reliable rather than due to evaluation noise, we conduct paired bootstrap resampling with 1000 samples. Table 4 reports the original baseline score, SSMoE score, absolute improvement, 95% confidence interval, and paired bootstrap $p$-value. Across both COCO and Flickr30k, all improvements are positive, with narrow confidence intervals and statistically significant $p$-values. On COCO, SSMoE improves R@1 by +0.8 points with a 95% confidence interval of $[0.20, 1.58]$ and $p = 0.002$. Similarly, on Flickr30k, SSMoE improves I2T@5 by +0.6 points with a confidence interval of $[0.40, 0.88]$ and $p < 0.001$. These results indicate that the gains of SSMoE are consistent and statistically significant despite their small absolute magnitude. Moreover, as shown in the corruption experiments, the improvements become larger under distribution shift, suggesting that the main benefit of SSMoE lies not only in improving saturated clean retrieval performance, but also in enhancing robustness through better expert utilization.

### 4.4. Ablation Studies

We investigate the effectiveness and robustness of SSMoE to the different hyper-parameter settings.

| Metric | CLIP-MoE | SSMoE (Ours) | $\Delta$ | 95% CI | $p$-value |
|---|---|---|---|---|---|
| **COCO I2T** | | | | | |
| @1 | 65.0 | 65.9 | +0.8 | [0.20, 1.58] | 0.002 |
| @5 | 86.0 | 86.5 | +0.5 | [0.00, 1.00] | 0.024 |
| @10 | 92.0 | 92.2 | +0.2 | [0.05, 0.40] | 0.018 |
| **Flickr30k I2T** | | | | | |
| @1 | 61.8 | 62.3 | +0.5 | [0.15, 0.80] | 0.001 |
| @5 | 83.4 | 84.1 | +0.6 | [0.40, 0.88] | < 0.001 |
| @10 | 89.6 | 89.9 | +0.3 | [0.06, 0.44] | 0.008 |

*Table 4.* Paired bootstrap significance testing on the clean setting using 1000 bootstrap samples. $\Delta$ denotes the absolute improvement of SSMoE over the CLIP-MoE baseline.

**Selective vs. averaged eigenvectors.** We next ablate whether SSMoE depends on the learned router for selecting eigenvector directions. In addition to the router-guided variant, we evaluate a router-free variant that simply averages the eigenvector descriptors of each expert. As shown in Table 5, this simple averaging strategy already improves over the original model by +3.9 points on average, indicating that the eigenvector representations themselves encode useful semantic information about expert specialization. Router-guided selection further improves the average gain to +5.3 points by filtering less relevant or misaligned directions. These results show that the router acts as a lightweight selection mechanism rather than a strict dependency: eigenvectors provide the core routing signal, while the router improves its precision.

**Number of Eigenvectors.** We assess the effectiveness and stability of SSMoE under varying hyper-parameter configurations. In particular, we examine the impact of changing the number of top eigenvectors (TOPC) used in the model. As shown in Table 6, performance across datasets demonstrates that SSMoE remains robust to different values of TOPC, with optimal results often achieved at higher settings. In practice, we observe that setting TOPC $\approx 50$ yields the best performance.

**Balancing factor $\alpha$.** We analyze the effect of the balancing factor $\alpha$ on the performance of OLMoE-1B across three

| Task | Orig. | SSMoE | SSMoE w/ Avg. EV |
|------|-------|-------|------------------|
| ARC-C | 44.2 | **52.1** | 49.4 |
| ARC-E | 77.9 | **82.4** | 80.7 |
| BoolQ | 62.8 | **72.4** | 69.8 |
| GSM8K | **36.8** | 30.8 | 33.3 |
| HellaSwag | 29.0 | **37.5** | 36.5 |
| OBQA | 19.8 | **23.0** | 22.8 |
| PIQA | 62.5 | **72.2** | 69.5 |
| WinoGrande | 57.5 | **62.2** | 59.6 |
| **Avg.** | 48.8 | **54.1** | 52.7 |

*Table 5.* Ablation of router-guided eigenvector selection. The router-free variant, which simply averages eigenvector descriptors, still outperforms the original model, while router-guided selection provides further gains.

classification tasks. As shown in Table 6, performance generally improves with increasing $\alpha$, with optimal results consistently observed around $\alpha = 0.9$. Empirically, we find that setting $\alpha$ in the range of approximately 0.5 to 0.9 provides a robust trade-off between the two, yields strong and stable performance.

| | TOPC | | | | $\alpha$ | | | | | |
|---------|------|------|------|------|------|------|------|------|------|------|
| Dataset | 5 | 10 | 20 | 50 | 0.0 | 0.3 | 0.5 | 0.7 | 0.9 | 1.0 |
| Emotion | 36.5 | 35.4 | 35.3 | **37.4** | 26.1 | 27.5 | 29.9 | 30.2 | **32.7** | 28.9 |
| Toxic | 57.4 | 57.6 | 60.0 | **60.7** | 60.2 | 61.1 | 59.8 | 62.2 | **62.6** | 59.3 |
| Tweet | 51.3 | 51.7 | 53.2 | **53.4** | 48.4 | 49.8 | 53.3 | 52.5 | **54.3** | 48.9 |

*Table 6.* Ablation studies on OLMoE-1B classification tasks. We vary the number of selected eigenvectors TOPC and the balancing factor $\alpha$. Higher values are better, with the best results highlighted in **bold**.

### 4.5. In-depth Analysis

> **Finding 4:** SMoE routers exhibit increased collapse in deeper layers, whereas EV routers maintain orthogonality.

**Collapse Problems.** In practice, we observe that the SMoE router exhibits increasing levels of collapse across layers, as shown in Figure 6a. In contrast, the EV router maintains consistent orthogonality layer by layer, as illustrated in Figure 6b, which is theoretically supported by Lemma B.1.

**Latent Structure Discovery.** Figure 7 presents t-SNE projections of two learned representations, colored by KMeans cluster assignments ($k = 64$). Visually, Representation 1 exhibits more coherent and well-separated clusters, while SMoE appears more dispersed with overlapping regions. This observation is quantitatively supported by clustering metrics: SSMoE achieves a higher Silhouette score (Rousseeuw, 1987) and a lower Davies-Bouldin score (Davies & Bouldin, 1979), indicating stronger intrinsic cluster structure. These results suggest that SSMoE

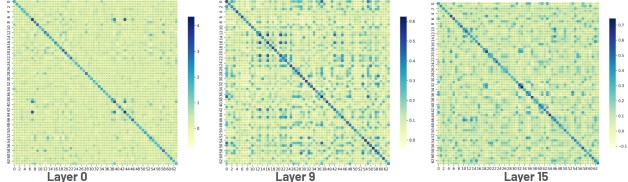

*(a)* Correlation between expert embeddings within the SMoE router of the OLMoE model.

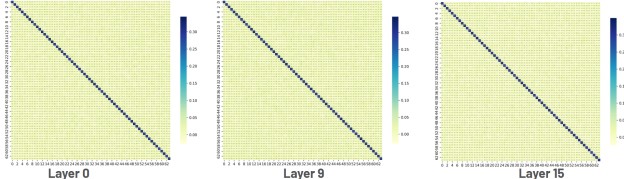

*(b)* Correlation between expert weight eigenvectors (EV router).

*Figure 6.* A comparison of collapse behavior shows that the OL-MoE router exhibits increased collapse at deeper layers, while the EV router does not encounter this issue.

better preserves semantically meaningful groupings in the latent space.

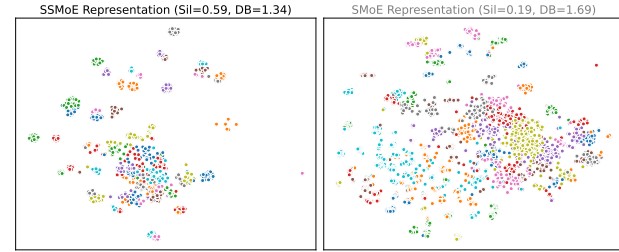

*Figure 7.* t-SNE visualizations of two learned representations, colored by KMeans cluster assignments with $k = 64$. SSMoE (left) displays more compact and well-separated clusters than SMoE (right), indicating stronger underlying cluster structure. Clustering quality is further supported by quantitative metrics: higher Silhouette and lower Davies-Bouldin index for SSMoE.

## 5. Conclusion

In this research, we investigate the behavior of routers in advanced Sparse Mixture of Experts (SMoE) models and highlight a common issue known as representation collapse. Building on this analysis, we propose *SVD-SMoE (SSMoE)*, a novel and efficient framework that enhances SMoE by leveraging *Eigenvalue Representations of Expert Weights*. SSMoE learns more robust representations and effectively addresses the representation collapse problem often encountered in traditional SMoEs. Experimental results across both training-free and fine-tuning settings show that SSMoE achieves more efficient and effective fine-tuning and inference than existing advanced SMoE models. We hope this work opens new directions for designing effective SMoE architectures without relying on conventional routers.

## Impact Statement

Our work aims to advance Machine Learning through transparent, reproducible research using publicly available resources. While the immediate scope of this study does not involve human participants, we remain mindful of the broader implications of LLM deployment. Specifically, models trained on uncurated web data are susceptible to systemic biases that require rigorous, continuous mitigation efforts. Additionally, we acknowledge that the high computational demands of LLM training pose sustainability challenges. This paper serves as a technical contribution while advocating for more resource-efficient and ethically-aware AI development.

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

# A. Supplementary Material for "Eigenvectors of Experts are Training-free Non-collapsing Routers"

The appendix is organized as follows: Section B presents the additional theoretical analysis, while Section C provides algorithm of the SSMoE. Supplementary experimental results are reported in Section D, and full implementation details are described in Section E.

# B. Additional Theoretical Analysis

**Lemma B.1** (Approximate orthogonality via Gaussian representation). *Let $\widetilde{\mathcal{C}}^{(1)}, \ldots, \widetilde{\mathcal{C}}^{(N)} \in \mathbb{R}^d$ be spectral embeddings of $N$ experts. Assume that for each $i$ the embedding $\widetilde{\mathcal{C}}^{(i)}$ is distributed approximately as a random unit vector (Thamm et al., 2022; Staats et al., 2026) drawn uniformly from the sphere $S^{d-1}$, and that different experts' embeddings are independent. Then for any fixed distinct indices $i \neq j$ and for every $\varepsilon > 0$,*

$$\mathbb{P}\big(|\widetilde{\mathcal{C}}^{(i)\top}\widetilde{\mathcal{C}}^{(j)}| \geq \varepsilon\big) \longrightarrow 0 \qquad \text{as } d \to \infty.$$

*In particular, $\widetilde{\mathcal{C}}^{(i)\top}\widetilde{\mathcal{C}}^{(j)} \xrightarrow{p} 0$, so different expert embeddings are (asymptotically) nearly orthogonal.*

Lemma B.1 establishes that the eigenvectors of expert weight matrices provide more effective routing signals than standard MoE routers, owing to their orthogonality properties. This theoretical insight is corroborated by the empirical results in Figure 6, which show that eigenvector-based routers exhibit near-orthogonality, whereas traditional routers suffer from high correlations.

## B.1. Proof for Lemma B.1

*Proof.* Represent independent standard normal vectors $X, Y \sim \mathcal{N}(0, I_d)$. A uniformly random unit vector on $S^{d-1}$ has the same distribution as $U := X/\|X\|$; similarly set $V := Y/\|Y\|$. Thus for two independent uniform unit vectors we may write their inner product as

$$\langle U, V \rangle = \frac{\langle X, Y \rangle}{\|X\| \, \|Y\|}.$$

By the strong law of large numbers,

$$\frac{\|X\|}{\sqrt{d}} = \sqrt{\frac{1}{d}\sum_{k=1}^{d} X_k^2} \xrightarrow{\text{a.s.}} 1, \qquad \frac{\|Y\|}{\sqrt{d}} \xrightarrow{\text{a.s.}} 1.$$

Moreover, $\langle X, Y \rangle = \sum_{k=1}^{d} X_k Y_k$ is a sum of i.i.d. mean-zero terms with variance 1, so by the central limit theorem

$$\frac{\langle X, Y \rangle}{\sqrt{d}} \xrightarrow{d} \mathcal{N}(0, 1).$$

Combining these two facts yields

$$\sqrt{d}\,\langle U, V \rangle = \frac{\langle X, Y \rangle/\sqrt{d}}{(\|X\|/\sqrt{d})(\|Y\|/\sqrt{d})} \xrightarrow{d} \mathcal{N}(0, 1),$$

hence for every fixed $\varepsilon > 0$,

$$\mathbb{P}\big(|\langle U, V \rangle| \geq \varepsilon\big) = \mathbb{P}\big(\sqrt{d}\,|\langle U, V \rangle| \geq \varepsilon\sqrt{d}\big) \leq \mathbb{P}\big(|\mathcal{N}(0, 1)| \geq \varepsilon\sqrt{d}/2\big) + o(1) \longrightarrow 0.$$

Therefore $\langle U, V \rangle \xrightarrow{p} 0$ as $d \to \infty$.

The lemma follows by taking $U = \widetilde{\mathcal{C}}^{(i)}$ and $V = \widetilde{\mathcal{C}}^{(j)}$ under the stated Random matrix theory (RMT) assumption that the spectral embeddings behave like independent uniform unit vectors. The same argument extends to any fixed finite collection of embeddings: all pairwise inner products vanish in probability as $d \to \infty$, so the collection is asymptotically orthogonal. □

### B.2. Proof for Lemma 3.1

*Proof.* Expand the product for $i \neq j$ and take expectation:

$$\mathbb{E}[\mathcal{V}_i \mathcal{V}_j] = \alpha^2 \mathbb{E}[f_{\text{EV},i} f_{\text{EV},j}] + (1-\alpha)^2 \mathbb{E}[s_i s_j] + \alpha(1-\alpha)\big(\mathbb{E}[f_{\text{EV},i} s_j] + \mathbb{E}[f_{\text{EV},j} s_i]\big).$$

By assumption (1) the first term is zero, and by assumption (2) the cross terms are zero. The remaining term is $(1-\alpha)^2 \mathbb{E}[s_i s_j]$, which proves the identity. The inequality and the monotonic dependence on $\alpha$ follow immediately because $(1-\alpha)^2 < 1$ for $\alpha \in (0,1]$. $\qquad\square$

**Extension to selective SSMoE.** The proof above corresponds to the router-independent case, such as the simple-average EV variant. For the router-guided selective variant, the learned router is used to select a subset of eigenvector directions. This selection can introduce mild dependence between $f_{\text{EV}}(x)$ and $s(x)$, so the exact decorrelation condition $\mathbb{E}[f_{\text{EV},i} s_j] = 0$ may not hold. We therefore consider the relaxed condition

$$|\mathbb{E}[f_{\text{EV},i} s_j]| \leq \varepsilon, \qquad \forall i, j,$$

for some small $\varepsilon \geq 0$.

Under this relaxed condition, using the same expansion as above gives

$$\mathbb{E}[\mathcal{V}_i \mathcal{V}_j] = \alpha^2 \mathbb{E}[f_{\text{EV},i} f_{\text{EV},j}] + (1-\alpha)^2 \mathbb{E}[s_i s_j] + \alpha(1-\alpha)\big(\mathbb{E}[f_{\text{EV},i} s_j] + \mathbb{E}[f_{\text{EV},j} s_i]\big).$$

By EV orthogonality,

$$\mathbb{E}[f_{\text{EV},i} f_{\text{EV},j}] = 0, \qquad i \neq j.$$

Taking absolute values and applying the triangle inequality, we obtain

$$|\mathbb{E}[\mathcal{V}_i \mathcal{V}_j]| \leq (1-\alpha)^2 \, |\mathbb{E}[s_i s_j]| + \alpha(1-\alpha) \, (|\mathbb{E}[f_{\text{EV},i} s_j]| + |\mathbb{E}[f_{\text{EV},j} s_i]|)$$
$$\leq (1-\alpha)^2 \, |\mathbb{E}[s_i s_j]| + 2\alpha(1-\alpha)\varepsilon.$$

Thus, selective SSMoE reduces pairwise router-logit correlation by the same $(1-\alpha)^2$ factor as in Lemma 2, up to an additional cross-correlation error term $2\alpha(1-\alpha)\varepsilon$. When $\varepsilon$ is small, the selective variant preserves the decorrelation effect while allowing the learned router to filter more relevant eigenvector directions.

## C. SSMoE Algorithm

Algorithm 1 summarizes the SSMoE layer. Importantly, eigenvectors are extracted from the expert Gram matrices rather than from the rectangular FFN weights directly, ensuring that the resulting spectral descriptors lie in the token representation space $\mathbb{R}^D$ and can therefore be used as routing projections.

## D. Additional Experiments

### D.1. Vision Tasks

For image-text retrieval, we conduct evaluations on the COCO (Lin et al., 2014) and Flickr30k (Plummer et al., 2015) datasets. For zero-shot image classification, we assess performance on six widely used benchmarks: CIFAR-10, CIFAR-100 (Krizhevsky, 2009), STL-10 (Coates et al., 2011), Caltech101 (Fei-Fei et al., 2004), ImageNet-1K (Deng et al., 2009), and ImageNet-O (Hendrycks et al., 2021b).

**Zero-Shot Image Classification Evaluation.** Table 7 presents the zero-shot image classification accuracy of CLIP-MoE and our proposed SSMoE across six widely used benchmarks under both clean and corrupted image conditions. Under the clean setting, SSMoE consistently outperforms CLIP-MoE on all datasets, achieving an average accuracy improvement of **+0.4%**. Notably, the largest gain is observed on Caltech101, where SSMoE improves accuracy by **+1.1%**.

In the corrupted setting, where input images are perturbed with random noise, SSMoE maintains its advantage, again outperforming CLIP-MoE across all benchmarks with an average improvement of **+0.4%**. The most significant gain is on ImageNet-O: **+0.7%**, which is known for its challenging out-of-distribution samples. These results demonstrate that SSMoE not only enhances performance in standard scenarios but also exhibits greater robustness under distributional shifts and noise.

---

**Algorithm 1** SSMoE Layer with Eigenvector-based Routing

---

**Input:** Tokens $X \in \mathbb{R}^{B \times L \times D}$; experts $\{E_i\}_{i=1}^{N}$; learned router logits $\mathcal{S}(X) \in \mathbb{R}^{B \times L \times N}$; FFN weights $W_i^{(1)}, W_i^{(2)}$; router embeddings $\{r_i\}_{i=1}^{N}$; Top-$c$, Top-$k$, mixing factor $\alpha \in [0, 1]$.
**Output:** $Y \in \mathbb{R}^{B \times L \times D}$.
**Pre-compute eigenvector descriptors.**
**for** $i = 1, \ldots, N$ **do**
$\quad A_i \leftarrow W_i^{(1)}(W_i^{(1)})^\top; \quad B_i \leftarrow (W_i^{(2)})^\top W_i^{(2)}$
$\quad \{v_j^{(i,1)}\} \leftarrow \text{EigVec}(A_i); \quad \{v_j^{(i,2)}\} \leftarrow \text{EigVec}(B_i)$
$\quad \mathcal{C}_1^{(i)} \leftarrow \text{TopC}(\{v_j^{(i,1)}\}, r_i, c)$
$\quad \mathcal{C}_2^{(i)} \leftarrow \text{TopC}(\{v_j^{(i,2)}\}, r_i, c)$
$\quad C_i \leftarrow \frac{1}{2}\left[\text{mean}(\mathcal{C}_1^{(i)}) + \text{mean}(\mathcal{C}_2^{(i)})\right]$
**end for**
**Build eigenvector router.**
$W_{\text{EV}} \leftarrow [C_1, \ldots, C_N] \in \mathbb{R}^{D \times N}$
**Compute routing probabilities.**
$F_{\text{EV}}(X) \leftarrow X W_{\text{EV}}$
$P_{\text{EV}}(X) \leftarrow \text{softmax}(F_{\text{EV}}(X))$
$P_{\text{R}}(X) \leftarrow \text{softmax}(\mathcal{S}(X))$
$P(X) \leftarrow \alpha P_{\text{EV}}(X) + (1 - \alpha)P_{\text{R}}(X)$
**Select experts and compute output.**
**for all** tokens $x_{b,\ell} \in X$ **do**
$\quad \mathcal{I}_{b,\ell} \leftarrow \text{TopK}(P(x_{b,\ell}), k)$
$\quad y_{b,\ell} \leftarrow \sum_{i \in \mathcal{I}_{b,\ell}} P_i(x_{b,\ell}) E_i(x_{b,\ell})$
**end for**
**return** $Y = \{y_{b,\ell}\}$

---

| Clean | | | |
|---|---|---|---|
| **Dataset** | **CLIP-MoE** | **SSMoE** | **Δ** |
| CIFAR-10 | 95.7 | **96.0** | **+0.3** |
| CIFAR-100 | 79.6 | **79.9** | **+0.3** |
| STL-10 | 99.2 | **99.3** | **+0.1** |
| Caltech101 | 83.6 | **84.7** | **+1.1** |
| ImageNet-1K | 74.6 | **74.9** | **+0.3** |
| ImageNet-O | 33.5 | **33.7** | **+0.2** |
| **Avg.** | 77.7 | **78.1** | **+0.4** |
| **Corrupt** | | | |
| **Dataset** | **CLIP-MoE** | **SSMoE** | **Δ** |
| CIFAR-10 | 66.7 | **66.9** | **+0.2** |
| CIFAR-100 | 34.4 | **34.9** | **+0.5** |
| STL-10 | 85.8 | **86.1** | **+0.3** |
| Caltech101 | 76.0 | **76.2** | **+0.2** |
| ImageNet-1K | 46.6 | **46.9** | **+0.3** |
| ImageNet-O | 31.0 | **31.7** | **+0.7** |
| **Avg.** | 56.8 | **57.1** | **+0.4** |

*Table 7.* Classification accuracy (%) of CLIP-MoE and SSMoE on multiple datasets. The "+/-" row indicates the improvement of SSMoE over CLIP-MoE. Best results per row are in **bold**.

## D.2. LLMs Finetuning Tasks

We showcase the effectiveness of our method through supervised fine-tuning on OLMoE-7B (Muennighoff et al., 2025).For training, we adopt Supervised Fine-Tuning (SFT) using GaLore (Zhao et al., 2024), which enables full-parameter optimization with improved memory efficiency compared to typical low-rank adaptation methods such as LoRA. The fine-tuning is conducted on the Alpaca dataset (Taori et al., 2023) for 2,000 steps, with evaluation performed every 10 steps.

**Reasoning Evaluation.** We evaluate the performance of OLMoE on six key reasoning benchmarks. The ARC-C and ARC-E benchmarks (Clark et al., 2018) assess scientific question answering at varying difficulty levels. BoolQ (Clark et al., 2019) evaluates reading comprehension through yes/no questions. PIQA (Bisk et al., 2020) measures a model's ability to perform physical commonsense reasoning by selecting the most plausible outcome in everyday situations. MMLU (Hendrycks et al., 2021a) tests broad general knowledge and reasoning capabilities, while WinoGrande (Sakaguchi et al., 2021) targets coreference resolution through challenging pronoun disambiguation tasks.

| Tasks | Method | ARC-C | ARC-E | BoolQ | Avg. |
|-------|--------|-------|-------|-------|------|
| | SMoE | **52.7** | 78.2 | 76.4 | 69.1 |
| OLMoE-7B | SSMoE | 52.3 | **80.9** | **77.5** | **70.2** |
| | $\Delta$ | -0.4 | +2.7 | +1.1 | +1.1 |

| Tasks | Method | PiQA | MMLU | WinoGrande | Avg. |
|-------|--------|------|------|------------|------|
| | SMoE | 81.2 | 50.8 | 69.9 | 67.3 |
| OLMoE-7B | SSMoE | **82.5** | **51.0** | **71.2** | **68.2** |
| | $\Delta$ | +1.3 | +0.2 | +1.3 | +0.9 |

*Table 8.* Performance comparison between SSMoE and the SMoE baseline on reasoning tasks. Higher is better, best results per row are in **bold**.

Overall, SSMoE outperforms the SMoE baseline on six benchmarks for the OLMoE model as Table 8. Notably, on ARC-E, PIQA, and WinoGrande, SSMoE achieves gains of **1–3%** over the baselines. These results suggest that SSMoE is not only effective in training-free settings for downstream tasks, but also excels at handling complex reasoning tasks.

## D.3. LLMs Embeding Evaluation

We present a comprehensive evaluation of three state-of-the-art SMoE models: **OLMoE-7B** (Table 9), **Qwen-MoE-7B** (Table 10), and **DeepSeekMoE-16B** (Table 11). Our results highlight the effectiveness of our method across different models and prompting conditions, demonstrating consistent improvements over baseline approaches such as the SMoE.

## D.4. Comparison with Training-free Routing

We compare SSMoE with two training-free routing baselines on GPT-OSS-20B under the 10-shot setting: hash-based routing and random routing. Hash-based routing assigns tokens to experts using a fixed token-dependent hash function, while random routing samples experts without using token–expert compatibility. As shown in Table 12, SSMoE consistently improves over both baselines on most tasks and achieves the best average performance. In particular, SSMoE improves the average score from 36.1 to 54.1 over hash-based routing, corresponding to an absolute gain of +18.0 points. Compared with random routing, SSMoE improves the average score from 45.0 to 54.1, yielding an absolute gain of +9.1 points. These results suggest that the proposed eigenvector-based routing captures meaningful expert specialization beyond input-agnostic or heuristic expert assignment strategies.

## D.5. Comparison on Distribution-Sensitive Metrics

To further assess whether the gains of SSMoE arise from meaningful improvements rather than uncontrolled changes in model behavior (Dutta et al., 2024), we report additional distribution-sensitive metrics, including answer flips and perplexity. These metrics are less directly optimizable than task accuracy and therefore provide complementary evidence about the stability and reliability of the proposed routing mechanism.

**Answer flips.** We first analyze prediction flips between the original model and SSMoE. In particular, we distinguish harmful flips, where a previously correct prediction becomes incorrect ($\mathbf{C}\rightarrow\mathbf{W}$), from beneficial flips, where a previously

incorrect prediction becomes correct ($\mathbf{W}{\to}\mathbf{C}$). As shown in Table 13, SSMoE produces more beneficial flips than harmful ones on average. Across eight benchmarks, SSMoE achieves an average $\mathbf{W}{\to}\mathbf{C}$ rate of $13.5\%$, compared to a $\mathbf{C}{\to}\mathbf{W}$ rate of $9.2\%$, yielding a net accuracy gain of $+5.3\%$. This indicates that the observed improvements are primarily driven by correcting errors rather than by random prediction changes.

**Perplexity.**    We also evaluate perplexity on WikiText-2 using GPT-OSS-20B. Perplexity provides a direct measure of whether routing changes introduce harmful distribution shift in language modeling behavior. As reported in Table 14, SSMoE achieves lower perplexity than both the original model and the Router baseline. This suggests that SSMoE improves compatibility with the data distribution rather than degrading general language modeling quality. We note that the absolute perplexity values are higher than those commonly reported for standard language-modeling benchmarks because GPT-OSS-20B is an instruction-tuned LLM. Therefore, the main comparison is the relative difference under the same evaluation setup. The lower perplexity of SSMoE compared with the Router baseline further indicates that the improvement is not merely caused by modifying the routing behavior, but by selecting experts more effectively.

### D.6. Correlation of Expert Selection between SSMoE and SMoE Routers

We further analyze whether SSMoE simply reproduces the expert choices of the original SMoE router or provides a complementary routing signal. To this end, we measure the mean overlap between the experts selected by the eigenvector-based routing mechanism and those selected by the original learned router. A high overlap would suggest that SSMoE mostly mimics the baseline router, whereas a low overlap indicates that the eigenvector descriptors induce substantially different expert selection patterns.

Table 15 reports the mean overlap across eight benchmarks. SSMoE exhibits consistently low overlap with the original router, with an average overlap of only $0.1531$. This shows that the eigenvector-based routing mechanism does not trivially replicate the original router's decisions. Instead, it selects a different subset of experts, suggesting that the eigenvector descriptors capture complementary information about expert specialization. This complementary selection signal helps explain why SSMoE can improve performance over the original routing mechanism.

### D.7. Effect of Expert Capacity on Mathematical Reasoning

We further investigate the performance drop observed on GSM8K under the pruned expert setting. Our analysis suggests that this drop is mainly caused by reducing the available expert pool, which can remove task-critical experts for reasoning-heavy domains such as mathematics. Unlike some knowledge or commonsense benchmarks, mathematical reasoning often relies on a broader set of specialized computational and symbolic capabilities. Therefore, pruning experts may disproportionately affect tasks such as GSM8K by excluding experts that are important for multi-step arithmetic reasoning.

To verify this, we compare SSMoE under two settings: using the full expert pool and using only $75\%$ of the experts. As shown in Table 16, SSMoE with full expert capacity substantially outperforms both the original model and the pruned SSMoE variant. Specifically, full-capacity SSMoE improves over the original model by $+6.6$ points and over $75\%$ of the experts by $+12.6$ points. These results indicate that the degradation on GSM8K is not due to the eigenvector-based routing mechanism itself, but rather to the loss of important experts caused by pruning. This finding suggests that expert capacity is particularly important for mathematical reasoning tasks. While SSMoE can improve expert selection, aggressive expert pruning may remove experts that are necessary for solving reasoning-intensive problems. Thus, for mathematics-heavy benchmarks, preserving the full expert pool is preferable when the goal is to maximize accuracy.

## E. Implementation Details

For the Large Reasoning Model experiments, we evaluate our method using the lm-evaluation-harness library (Gao et al., 2024). All experiments with the GPT-OSS models (OpenAI et al., 2025) are performed on two H200 GPUs.

For the Large Language Models experiments, we implement our method based on the publicly available MoEE implementation (Li & Zhou, 2025)[1]. Due to resource limitations, both our method and the baselines are evaluated using 4-bit quantization with a batch size of 128. Experiments with the *OLMoE-7B* model are conducted on a single H100 GPU, whereas the *Qwen-MoE-7B* and *DeepSeekMoE-16B* models are tested using two H100 GPUs.

---

[1]https://github.com/tianyi-lab/MoE-Embedding

For Vision Language models, we implement SSMoE as a modular extension to the publicly available CLIP-MoE framework (Zhang et al., 2025)[2]. All experiments involving vision-language tasks are conducted using a single NVIDIA H100 GPU.

---

[2]https://github.com/OpenSparseLLMs/CLIP-MoE

| Category | Model | Dataset | Setting | Router | SMoE | MoEE | EV | SSMoE |
|---|---|---|---|---|---|---|---|---|
| Classification | OLMoE-7B | Emotion | None | 24.1 | 24.5 | 25.1 | 32.3 | **37.4** |
| | | | Prompt | 27.6 | 49.9 | 44.5 | 29.9 | **53.5** |
| | | Toxic | None | 51.9 | 58.9 | 51.9 | 52.5 | **60.8** |
| | | | Prompt | 52.3 | 65.2 | 53.4 | 52.1 | **67.4** |
| | | Tweet | None | 47.7 | 46.8 | 48.4 | 53.0 | **53.5** |
| | | | Prompt | 49.5 | 58.0 | 57.2 | 48.7 | **60.9** |
| Clustering | OLMoE-7B | Medrxiv | None | 15.0 | 17.6 | 17.4 | 15.5 | **19.0** |
| | | | Prompt | 15.8 | 23.9 | 22.0 | 14.7 | **27.5** |
| | | 20Groups | None | 12.4 | 11.8 | 11.5 | 12.9 | **13.7** |
| | | | Prompt | 16.7 | 25.7 | 24.4 | 12.5 | **32.7** |
| Pair Classification | OLMoE-7B | SemEval | None | 43.6 | 35.8 | 43.6 | **50.7** | 44.8 |
| | | | Prompt | 45.7 | 46.7 | 53.8 | 47.2 | **55.7** |
| | | URLCorpus | None | 47.0 | 42.4 | 47.8 | **67.7** | 57.5 |
| | | | Prompt | 61.4 | 77.4 | 78.2 | 72.4 | **80.5** |
| Reranking | OLMoE-7B | Ask | None | 41.3 | 41.0 | 41.4 | **43.4** | 42.2 |
| | | | Prompt | 43.4 | **51.9** | 50.2 | 44.0 | 51.5 |
| | | SciDocs | None | 45.5 | 46.3 | 50.8 | 49.6 | **57.4** |
| | | | Prompt | 53.6 | 69.6 | 75.1 | 50.5 | **76.8** |
| | | StackOver | None | 25.8 | 24.8 | 26.4 | **30.5** | 27.5 |
| | | | Prompt | 28.1 | 32.5 | 34.3 | 30.9 | **34.5** |
| STS | OLMoE-7B | Biosses | None | 39.3 | 13.6 | 29.7 | **65.2** | 36.1 |
| | | | Prompt | 51.2 | 61.8 | 70.2 | 53.8 | **75.2** |
| | | SickR | None | 50.3 | 46.3 | 53.0 | **55.7** | 55.6 |
| | | | Prompt | 51.9 | 65.7 | 66.1 | 49.0 | **66.8** |
| | | STS12 | None | 40.1 | 8.6 | 37.8 | **57.2** | 46.7 |
| | | | Prompt | 51.3 | 53.8 | 63.6 | 49.7 | **65.7** |
| | | STS13 | None | 40.5 | 21.1 | 43.4 | **56.1** | 52.8 |
| | | | Prompt | 52.5 | 66.5 | 72.7 | 53.9 | **75.5** |
| | | STS14 | None | 29.5 | 13.4 | 31.7 | **46.4** | 40.3 |
| | | | Prompt | 41.1 | 56.8 | 64.2 | 42.9 | **66.6** |
| | | STS15 | None | 30.8 | 27.8 | 33.3 | **51.6** | 43.1 |
| | | | Prompt | 46.4 | **69.3** | 66.4 | 47.4 | 67.1 |
| | | STS16 | None | 46.5 | 38.9 | 45.8 | **57.5** | 52.1 |
| | | | Prompt | 52.4 | **70.1** | 68.3 | 51.0 | **69.5** |
| | | STSBen | None | 42.2 | 23.4 | 44.5 | **52.3** | 49.5 |
| | | | Prompt | 48.6 | 63.6 | 70.7 | 41.8 | **71.8** |
| Summarization | OLMoE-7B | Medrxiv | None | 28.4 | 20.9 | 29.8 | 30.0 | **31.0** |
| | | | Prompt | 25.6 | 28.9 | **30.4** | 27.0 | 30.0 |

*Table 9.* Performance comparison of Router, SMoE, MoEE, SSMoE, EV, and SSMoE across MTEB Tasks with *OLMoE-7B* models. The best result for each row is highlighted in **bold**.

| Category | Model | Dataset | Setting | Router | SMoE | MoEE | EV | SSMoE |
|---|---|---|---|---|---|---|---|---|
| Classification | Qwen-MoE-7B | Emotion | None | 27.2 | 33.9 | 34.3 | 24.4 | **35.3** |
| | | | Prompt | 37.0 | 48.5 | 47.2 | 38.6 | **51.0** |
| | | Toxic | None | 53.0 | 61.1 | 52.9 | 52.9 | **61.9** |
| | | | Prompt | 53.4 | 64.5 | 54.1 | 53.8 | **65.3** |
| | | Tweet | None | 51.1 | 55.9 | 55.9 | 52.9 | **57.6** |
| | | | Prompt | 56.1 | 61.1 | 60.7 | 58.8 | **62.3** |
| Clustering | Qwen-MoE-7B | Medrxiv | None | 15.3 | 23.3 | 23.0 | 15.2 | **26.0** |
| | | | Prompt | 14.2 | 24.6 | 21.8 | 15.3 | **30.0** |
| | | 20Groups | None | 12.0 | **31.5** | 27.4 | 12.2 | 36.1 |
| | | | Prompt | 14.4 | 43.8 | 38.4 | 17.3 | **49.1** |
| Pair Classification | Qwen-MoE-7B | SemEval | None | 42.0 | 38.8 | 42.5 | 43.1 | **46.6** |
| | | | Prompt | 47.0 | 52.4 | 52.4 | 49.9 | **56.2** |
| | | URLCorpus | None | 49.8 | 54.9 | 60.6 | 50.8 | **64.1** |
| | | | Prompt | 56.7 | 68.7 | 68.2 | 61.2 | **74.3** |
| Reranking | Qwen-MoE-7B | Ask | None | 43.1 | 45.8 | 47.3 | 43.1 | **48.8** |
| | | | Prompt | 43.3 | 48.3 | 49.5 | 44.4 | **51.4** |
| | | SciDocs | None | 49.6 | 60.6 | **67.0** | 50.9 | 66.0 |
| | | | Prompt | 50.9 | 60.1 | 68.7 | 52.2 | 65.8 |
| | | StackOver | None | 26.2 | 29.5 | 31.1 | 26.9 | **32.6** |
| | | | Prompt | 28.8 | 31.3 | 35.2 | 29.6 | **35.4** |
| STS | Qwen-MoE-7B | Biosses | None | 33.8 | 32.5 | 49.6 | 40.0 | **65.6** |
| | | | Prompt | 55.1 | 55.8 | **68.4** | 58.2 | 67.7 |
| | | SickR | None | 51.0 | 55.5 | 61.0 | 51.2 | **62.1** |
| | | | Prompt | 50.2 | 59.7 | 64.3 | 52.0 | **67.2** |
| | | STS12 | None | 40.2 | 16.9 | 46.3 | 41.9 | **48.6** |
| | | | Prompt | 49.3 | 25.0 | 59.2 | 51.8 | **61.8** |
| | | STS13 | None | 38.1 | 42.9 | 56.7 | 41.1 | **64.2** |
| | | | Prompt | 53.3 | 57.5 | 73.4 | 57.2 | **75.3** |
| | | STS14 | None | 28.1 | 26.5 | 45.4 | 30.8 | **51.3** |
| | | | Prompt | 40.4 | 38.8 | 60.0 | 44.4 | **64.4** |
| | | STS15 | None | 34.8 | 40.5 | 46.1 | 37.0 | **51.7** |
| | | | Prompt | 40.7 | 52.3 | 58.8 | 46.5 | **64.3** |
| | | STS16 | None | 47.6 | 51.0 | 58.1 | 49.8 | **62.0** |
| | | | Prompt | 51.6 | 64.2 | 65.7 | 53.3 | **67.6** |
| | | STSBen | None | 37.0 | 37.7 | 50.9 | 40.6 | **57.7** |
| | | | Prompt | 45.6 | 47.8 | 64.5 | 50.1 | **69.3** |
| Summarization | Qwen-MoE-7B | Medrxiv | None | 28.3 | 13.4 | **31.2** | 30.3 | 29.6 |
| | | | Prompt | 27.0 | 23.0 | 27.3 | 26.4 | **30.7** |

*Table 10.* Performance comparison of Router, SMoE, MoEE, SSMoE, EV, and SSMoE across MTEB Tasks with *Qwen-MoE-7B* models. The best result for each row is highlighted in **bold**.

| Category | Model | Dataset | Setting | Router | SMoE | MoEE | EV | SSMoE |
|---|---|---|---|---|---|---|---|---|
| Classification | DeepSeekMoE-16B | Emotion | None | 26.1 | 27.4 | 27.6 | 27.1 | **31.6** |
| | | | Prompt | 37.9 | 48.3 | 46.4 | 38.1 | **48.9** |
| | | Toxic | None | 53.3 | **60.4** | 53.1 | 52.7 | 59.0 |
| | | | Prompt | 53.1 | 62.4 | 53.6 | 54.3 | **67.0** |
| | | Tweet | None | 51.0 | 51.9 | 52.6 | 52.1 | **55.2** |
| | | | Prompt | 54.9 | 58.4 | 58.9 | 57.1 | **60.9** |
| Clustering | DeepSeekMoE-16B | Medrxiv | None | 15.1 | 23.0 | 22.0 | 15.8 | **24.3** |
| | | | Prompt | 17.0 | 25.7 | 24.0 | 17.0 | **27.4** |
| | | 20Groups | None | 11.7 | 13.2 | 13.7 | 12.8 | **18.9** |
| | | | Prompt | 18.6 | 32.3 | 33.0 | 20.2 | **35.1** |
| Pair Classification | DeepSeekMoE-16B | SemEval | None | 44.6 | 40.2 | 43.5 | 44.3 | **41.8** |
| | | | Prompt | 48.4 | 47.2 | 51.3 | 48.8 | **53.4** |
| | | URLCorpus | None | 46.4 | 41.7 | 48.6 | 49.1 | **60.6** |
| | | | Prompt | 66.5 | 72.4 | 75.4 | 69.7 | **75.6** |
| Reranking | DeepSeekMoE-16B | Ask | None | 41.7 | 41.1 | 42.3 | 41.4 | **43.1** |
| | | | Prompt | 43.5 | 43.8 | 46.9 | 44.9 | **50.4** |
| | | SciDocs | None | 48.2 | 50.6 | 57.1 | 49.7 | **61.7** |
| | | | Prompt | 58.3 | 65.6 | 72.6 | 60.1 | **72.6** |
| | | StackOver | None | 25.7 | 24.9 | 27.3 | 26.5 | **30.1** |
| | | | Prompt | 29.7 | 27.6 | 32.3 | 30.6 | **33.5** |
| STS | DeepSeekMoE-16B | Biosses | None | 29.5 | 31.7 | 26.8 | 30.7 | **54.4** |
| | | | Prompt | 47.0 | 40.1 | 57.6 | 48.8 | **66.0** |
| | | SickR | None | 50.4 | 47.4 | 53.1 | 52.4 | **57.8** |
| | | | Prompt | 56.0 | 61.9 | 65.8 | 58.2 | **66.8** |
| | | STS12 | None | 44.0 | 4.3 | 45.0 | 45.6 | **46.7** |
| | | | Prompt | 57.8 | 31.0 | 64.0 | 60.1 | **64.7** |
| | | STS13 | None | 36.0 | 28.4 | 41.1 | 38.0 | **50.0** |
| | | | Prompt | 55.3 | 56.0 | 70.9 | 60.8 | **74.3** |
| | | STS14 | None | 25.4 | 12.0 | 28.2 | 28.3 | **40.3** |
| | | | Prompt | 44.9 | 41.0 | 58.6 | 49.8 | **64.0** |
| | | STS15 | None | 34.8 | 33.9 | 38.7 | 35.8 | **44.9** |
| | | | Prompt | 49.7 | 46.5 | 58.5 | 54.3 | **64.1** |
| | | STS16 | None | 44.9 | 34.4 | 46.9 | 49.1 | **55.1** |
| | | | Prompt | 56.7 | 58.0 | 64.5 | 61.9 | **67.4** |
| | | STSBen | None | 36.6 | 18.3 | 42.1 | 40.0 | **51.8** |
| | | | Prompt | 54.9 | 57.7 | 67.8 | 60.0 | **71.0** |
| Summarization | DeepSeekMoE-16B | Medrxiv | None | 24.9 | 22.0 | 24.4 | 28.9 | **29.9** |
| | | | Prompt | 29.1 | 24.4 | 29.2 | 28.8 | **29.6** |

*Table 11.* Performance comparison of Router, SMoE, MoEE, SSMoE, EV, and SSMoE across MTEB Tasks with *DeepSeekMoE-16B* models. The best result for each row is highlighted in **bold**.

| Task | Hash Router | Random Router | SSMoE (Ours) |
|------|-------------|---------------|--------------|
| ARC-C | 23.3 | 33.5 | **52.1** |
| ARC-E | 36.7 | 65.8 | **82.4** |
| BoolQ | 59.3 | 66.5 | **72.4** |
| GSM8K | 14.4 | **33.1** | 30.8 |
| HellaSwag | 29.2 | 29.7 | **37.5** |
| OBQA | **26.2** | 20.6 | 23.0 |
| PIQA | 53.2 | 57.2 | **72.2** |
| WinoGrande | 46.9 | 53.3 | **62.2** |
| Average | 36.1 | 45.0 | **54.1** |

*Table 12.* Comparison with hash-based and random routing on GPT-OSS-20B under the 10-shot setting. SSMoE achieves the best average performance and substantially outperforms hash-based routing.

| Task | Original | SSMoE | Gap | FlipRate | C→W | W→C |
|------|----------|-------|-----|----------|-----|-----|
| ARC-C | 44.2 | 52.1 | +7.9 | 15.7 | 7.3 | 8.5 |
| ARC-E | 77.9 | 82.4 | +4.5 | 8.7 | 3.4 | 5.3 |
| BoolQ | 62.8 | 72.4 | +9.6 | 39.2 | 12.6 | 26.7 |
| GSM8K | 36.8 | 30.8 | -6.0 | 39.6 | 25.6 | 14.0 |
| HellaSwag | 29.0 | 37.5 | +8.5 | 12.1 | 2.4 | 9.7 |
| OBQA | 19.8 | 23.0 | +3.2 | 12.0 | 3.4 | 8.6 |
| PIQA | 62.5 | 72.2 | +9.7 | 21.6 | 5.1 | 16.4 |
| WinoGrande | 57.5 | 62.2 | +4.7 | 32.9 | 13.8 | 19.1 |
| **Avg.** | **48.8** | **54.1** | **+5.3** | **22.7** | **9.2** | **13.5** |

*Table 13.* Flip analysis between the original model and SSMoE. **C→W** denotes harmful flips from correct to wrong predictions, while **W→C** denotes beneficial flips from wrong to correct predictions. SSMoE produces more beneficial than harmful flips on average.

| Data | Model | Method | PPL ↓ |
|------|-------|--------|-------|
| WikiText-2 | GPT-OSS-20B | Original | 82.12 |
| | | Router | 75.91 |
| | | SSMoE | **73.23** |

*Table 14.* Perplexity comparison on WikiText-2 using GPT-OSS-20B. Lower perplexity is better. SSMoE reduces perplexity compared with both the original model and the Router baseline.

| Task | Mean Overlap |
|------|--------------|
| ARC-C | 0.1581 |
| ARC-E | 0.1586 |
| BoolQ | 0.1526 |
| GSM8K | 0.1581 |
| HellaSwag | 0.1488 |
| OBQA | 0.1462 |
| PIQA | 0.1540 |
| WinoGrande | 0.1486 |
| **Mean** | **0.1531** |

*Table 15.* Mean overlap between experts selected by SSMoE's eigenvector-based routing and the original SMoE router. Lower overlap indicates that SSMoE selects substantially different experts rather than simply mimicking the learned router.

| Task | Orig. | SSMoE (Full) | SSMoE (75% Experts) |
|------|-------|--------------|---------------------|
| GSM8K | 36.8 | **43.4** | 30.8 |

*Table 16.* Effect of expert capacity on GSM8K using GPT-OSS-20B. SSMoE performs strongly when the full expert pool is preserved, while pruning to 75% experts causes a large degradation on this reasoning-heavy task.

