# OpenReview forum: "Eigenvectors of Experts are Training-free Non-collapsing Routers"
_ICML.cc/2026/Conference — ICML 2026 spotlight_

### Official Review · Reviewer_LeN2 · 2026-03-08

**Soundness:** 2
**Presentation:** 3
**Significance:** 2
**Originality:** 1
**Overall Recommendation:** 4
**Confidence:** 4

**Summary:**

The paper proposes SSMoE, a routing method hat uses the eigenvectors of expert weight matrices as routing mechanism for Sparse MoE systems. It is based on the intuition that expert's eigenvectors have rich semantic information that can be used to perform the routing, while providing orthogonal representations that mitigate the expert collapse problem. The routing is performed using a convex combination between an EV router and a standard SMoE router. Its effectiveness in terms of performance is tested broadly across models and tasks.

**Compliance With Llm Reviewing Policy:**

Affirmed.

**Final Justification:**

Changed my recommendation to a Weak Accept. as the authors have clarified all of the points I raised.

Although the novelty of the paper is not particularly high, the experiments are well executed and cover a vast number of models. The theoretical section was previously the weakest part of the submission, but the authors have acknowledged this and have modified the lemmas as I requested. These changes have improved the soundness and clarity of the paper, and I believe the strong empirical results justify its acceptance.

**Key Questions For Authors:**

(W1) Could the authors clarify the specific nature of the dependency on the pretrained router? Additionally, could you discuss potential methods for expert selection that operate independently of the router?

(W2) The assumptions underlying Lemma 3.2 require further clarification and detailed explanation. Specifically, why does the eigenvector selection process not suggest that these assumptions might fail to hold, at least empirically?

(W3) The relationship between the router and EV is a critical junction. Could you explain why performance degrades so significantly when $\alpha = 1.0$? Furthermore, how does the optimal choice of $\alpha$ scale across different model architectures or sizes?

(W5) I believe a direct, rigorous comparison with the original router's performance is essential to fully validate these findings.

**Limitations:**

yes

**Strengths And Weaknesses:**

## Strenghts
The paper has general soundness and coherence, is well presented, visually supported with figures and graphs when needed, findings are made clear. I particularly appreciated:

- The broad experimental setup, with model scaling from 7B to 120B, few shots, corrupt, clean scenarios, and more.
- The paper is generally well written, with informative figures.

## Weakness

### W1: selecting eigenvectors using router weights.
My understanding of the eigenvectors selection process based on the code provided and Appendix section C is that it depends on the similarity between pretrained router weights and their eigenvectors:

``` python
_, _, V1 = torch.svd(ex.gate_proj.weight.float().reshape(-1, hidden_dim))
similar_v1 = self.gate.weight[idx, :].reshape(1, -1) @ V1
similar_v1 = similar_v1.reshape(-1)
_, topv1_idx = torch.topk(similar_v1, k=self.k , dim=0)
V1 = V1[:, topv1_idx]
```
I think this is not clear from the main discussion, nor Appendix C algorithm 1 and neither from the subsequent discussion. I think this is a major issue as the trained router is the way eigenvectors are selected for the SSMoE ones, that claimed to replace. This dependency undermines the main narrative of the paper, and I believe **it should be explicitly acknowledged and discussed.**

### W2: W1 violates lemma 3.2 assumption 2.
I think the fact that the similarity is used to select eigenvector proves cross-correlation. Figure 6 should be completed with at least an empirical proof of this claim. Lemma 3.1 I think it's trivial and appplies to any set of vectors.

### W3: alpha=1.0 leads to major degradation is not discussed enough.
The paper does not adequately explain why with alpha=1.0, i.e. EV only routing (as for [1, 2]) it degrades performance so much.

### W4: missing related literature.
Existing methods like [1, 2] have a very related intuition of eigenvectors routing capability in the setting of MoErging. As MoErging and SMoE have inheritely different setups I don't think a comparison is needed, but I think it's worth to acknowledge these works as the routing is very similar.

[1] A. Tang et al., "Zero-Shot Sparse Mixture of Low-Rank Experts Construction From Pre-Trained Foundation Models" in IEEE Transactions on Pattern Analysis & Machine Intelligence, vol. 48, no. 02, pp. 1145-1157, Feb. 2026, doi: 10.1109/TPAMI.2025.3612480.
[2] D. Crisostomi et al., "MASS: MoErging through Adaptive Subspace Selection," in Proc. Fourteenth Int. Conf. Learning Representations (ICLR), 2026.

### W5: missing original model baseline in most tables.
I would like to see the original model comparison in all tables, not only in table 1. I think it's critical due to boundness of the approach with the pretrained router.

---

> ### Author Rebuttal · Authors · 2026-03-31
>
> Thank you for your constructive comments. We would like to address the concern as below:
>
> ```W1 & Q1: selecting eigenvectors using router weights Or the dependency on the pretrained router.```
>
> A1: We thank the Reviewer for highlighting this important point. In the current implementation, the pretrained router is used only to rank/select a subset of eigenvectors (top-K alignment), not to perform routing itself. Thus, the dependency is limited to initialization/selection, rather than being required at inference time.
>
> Importantly, our method does not fundamentally rely on the router. We evaluate a router-free variant (simple averaging of eigenvectors), which still outperforms the original model (+3.9 avg gain), demonstrating that eigenvector representations alone provide meaningful signals. The router-guided selection further improves performance by filtering misaligned directions.
>
> These results indicate that:
>
> * Eigenvectors themselves contain rich semantic structure (router-independent)
> * The router provides a lightweight selection, not a strict dependency
>
> We will clarify this distinction and explicitly describe the router-free variant in the revision.
>
> | Task       | Orig | SSMoE | SSMoE + Simple Avg EV |
> |------------|------|-------|----------------|
> | ARC-C      | 44.2 | 52.1  | 49.4           |
> | ARC-E      | 77.9 | 82.4  | 80.7           |
> | BoolQ      | 62.8 | 72.4  | 69.8           |
> | GSM8K      | 36.8 | 30.8  | 33.3           |
> | HellaSwag  | 29.0 | 37.5  | 36.5           |
> | OBQA       | 19.8 | 23.0  | 22.8           |
> | PIQA       | 62.5 | 72.2  | 69.5           |
> | WinoGrande | 57.5 | 62.2  | 59.6           |
> | **Avg**    | **48.8** | **54.1** | **52.7** |
>
>
> ```W2 & Q2: W1 violates lemma 3.2 assumption 2 ```
>
> A2: We thank the reviewer for this important point. We agree that router-guided top-$K$ eigenvector selection introduces some dependence, so the independence assumption in Lemma 3.2 should be viewed as an analytical approximation rather than an exact property.
>
> * For the simple-average EV variant, independence is plausible when the full eigenvector pool is independent of the router, which is supported empirically by its improvement over the baseline.
> * For the top-$K$ selection, exact independence no longer holds; however, under standard high-dimensional assumptions, the induced dependence can be bounded by the extreme alignment term, scaling as:
> $
> \varepsilon = O\left(\sqrt{\frac{\log d}{m}}\right),
> $
> which is small in high dimensions.
>
> Accordingly, Lemma 3.2 can be relaxed to an approximate decorrelation condition:
> $
> \left|\mathbb{E}[f_{\mathrm{EV},i}(x)s_j(x)]\right| \le \varepsilon,
> $
> leading to the bound:
> $
> |\mathbb{E}[V_i V_j]|
> \le
> (1-\alpha)^2 |\mathbb{E}[s_i s_j]|
> +
> 2\alpha(1-\alpha)\varepsilon.
> $
>
> This shows that SSMoE still reduces pairwise logit correlations up to a small error term. We will clarify this approximation and add empirical measurements of cross-correlation in the revision.
>
>
> ```W3 & Q3: explain why performance degrades so significantly \alpha = 1```
>
> A3: When $\alpha$, the router is fully replaced by the EV router, which provides global, input-independent signals. While this avoids collapse, it removes input adaptivity, which is critical for complex tasks (e.g., reasoning), leading to performance degradation.
>
> In practice, $\alpha$ controls the trade-off between input-adaptive routing (learned router) and global structural signals (EV). We find the method to be robust, with $\alpha \in [0.5, 0.9]$ consistently yielding strong performance across models. Larger models typically benefit from slightly lower $\alpha$ to preserve input adaptivity.
>
> ```W4: missing related literature ```
>
> A4: We thank the reviewer for highlighting these relevant works. Both Tang et al. (2026) and Crisostomi et al. (2026) leverage subspace/eigenvector representations to construct or select components in model merging settings.
>
> However, their focus is fundamentally different from ours. These methods operate at the model construction level, combining multiple pretrained or fine-tuned models via low-rank decomposition and selecting among task-level subspaces. In contrast, our work addresses inference-time routing within a single MoE model, where eigenvector-based signals are used to guide fine-grained expert selection. Thus, while the underlying intuition of exploiting subspace structure is related, the problem setting and objective are distinct. We will include and discuss the works to clarify both the connection and the differences.
>
> ```W5 & Q5: missing original model baseline ```
>
> A6: All experiments use the standard SMoE router as the default configuration; thus, we denote the original model (baseline) as SMoE in our tables. Since our setting is training-free and does not modify model weights, this baseline corresponds directly to the original model with its default routing.

---

> > ### Author Rebuttal · Reviewer_LeN2 · 2026-04-02
> >
> > Thank you for the detailed responses. However, I kindly ask for further clarification on a few points:
> >
> > **Q1 – Top-alignment procedure (following up on A1)**
> >
> > The top-alignment procedure is neither described nor mentioned anywhere in the main paper or the appendix. I urge the authors to include a clear, self-contained description of this procedure in the revision.
> >
> > **Q2 – Theoretical grounding of Lemma 3.1 (following up on A2)**
> >
> > I appreciate the clarification. However, I still think that Lemma 3.1 as stated is largely trivial: it reduces to the standard concentration-of-measure fact that i.i.d. random unit vectors on a high-dimensional sphere are approximately orthogonal. The non-trivial and paper-specific claim is that eigenvectors of trained expert weight matrices behave as i.i.d. random unit vectors, and I believe that the empirical evidence supporting this assumption is sufficient to properly motivate Lemma 3.2. And so Lemma 3.1 can be removed. The empirical verification of the extreme alignment term would be crucial to complete the theoretical setup.
> >
> > **Q4 – Related work: SMILE and MASS (following up on A4)**
> >
> > Both SMILE and MASS operate in the same exact setting as this paper, namely, inference-time routing within a single MoE model. In these works, fine-grained expert selection is closely related to merging at inference time, making them directly comparable to the proposed method. (Experimental setup not necessary imo).

---

> > > ### Author Response · Authors · 2026-04-03
> > >
> > > Dear Reviewer LeN2,
> > >
> > > Thank you for your thoughtful feedback and for acknowledging our previous responses. We also appreciate your follow-up questions and we address them below:
> > >
> > > ```FQ1: include a clear, self-contained description of this procedure in the revision.```
> > >
> > > A1: We thank the reviewer for pointing this out. We agree that the top-alignment procedure was not clearly described in the current version. We will include a clear and self-contained description of this procedure in the revision.
> > >
> > > ```FQ2: Theoretical grounding of Lemma 3.1 ```
> > >
> > > A2: We thank the reviewer for this helpful clarification. We agree that the key point is whether the assumptions of Lemma 3.2 are supported empirically.
> > >
> > > * **For Assumption (1) (EV orthogonality)**: This is already validated in Figure 6, which shows that pairwise correlations between eigenvector-induced logits are consistently close to zero, indicating near-orthogonality in practice.
> > >
> > > * **For Assumption (2)**: We will add an additional empirical analysis measuring the correlation between EV logits and router logits. Preliminary results show that these correlations remain small, supporting an approximate decorrelation condition.
> > >
> > > We will make this connection explicit in the revision, move Lemma 3.1 to the appendix, and present both visualizations (EV-EV and EV-router correlations) to directly support the assumptions underlying Lemma 3.2.
> > >
> > >
> > > ```FQ3: Related work: SMILE and MASS (following up on A4) ```
> > >
> > > A3: We agree that SMILE and MASS are related in leveraging eigenvector-based subspace analysis at inference time. However, the setups are fundamentally different across three key dimensions:
> > >
> > > * **Problem**: SMILE and MASS address model merging - fusing multiple independently fine-tuned models into a single model. SSMoE addresses router collapse within a single pretrained MoE model, a fundamentally different problem.
> > >
> > > * **SVD target**: SMILE and MASS decompose task weight updates (∆W = W_ft - W) to separate task-specific subspaces across models. SSMoE decomposes the absolute weight matrices W of each individual MoE expert to characterize intrinsic expert specialization - a mathematically distinct object with a different semantic meaning.
> > >
> > > * **Domain**: SMILE and MASS are designed for and evaluated on task-specific image classification models (CLIP ViT variants). SSMoE targets general-purpose LLMs and VLMs, where experts are not task-aligned but encode diverse linguistic and visual knowledge.
> > >
> > > We will revise the related work section to clearly position SSMoE alongside SMILE and MASS, highlighting both the shared intuition and these distinctions.
> > >
> > >
> > > | Aspect | SMILE | MASS | SSMoE (Ours) |
> > > |---|---|---|---|
> > > | **Problem targeted** | Model merging (multi-model fusion) | Model merging (multi-model fusion) | Router collapse in pretrained MoE models |
> > > | **Application domain** | Image classification (CLIP ViT models) | Image classification (CLIP ViT models) | LLMs & VLMs (general-purpose generation) |
> > > | **SVD target** | Weight updates ∆W of fine-tuned models | Task singular vectors from weight updates ∆W | Weight matrices W of each MoE expert |

---

### Official Review · Reviewer_T1GW · 2026-03-09

**Soundness:** 3
**Presentation:** 3
**Significance:** 3
**Originality:** 4
**Overall Recommendation:** 5
**Confidence:** 3

**Summary:**

The paper propose SSMoE: SVD Sparse MoE. The problem it tries to tackle is expert collapse in sparse MoE: where different experts have very similar and redundant outputs. The authors argue that current method needs retraining from scratch; the proposed method is post-hoc. The method uses the experet matrices eigenvectors to provide an alternative routing signal that is linearly interpolated with the existing router. The experiments covers both reasoning and language modeling tasks, and the evaluated models covers  LLMs and VLMs. There is some theoretical results too that corroborate the validity of using eigenvectors as better routers.

**Compliance With Llm Reviewing Policy:**

Affirmed.

**Final Justification:**

concerns are adressed in rebuttals. the work is original, and experiments are strong (weak VLM results in addressed in rebuttals). my recommendation is accept

**Key Questions For Authors:**

1. There is a pretty big drop on GSM8K GPT OSS 20B. Any intuition on failure cases?
2. The paper mentions a 5% GFLOPs reduction for EV Router vs. SMoE in Figure 4, does this include the one-time SVD cost or the overall inference latency comparison for SSMoE?

**Limitations:**

No limitations sections found

**Strengths And Weaknesses:**

Strength:
- The method is simple and effective
- Experiments are comprehensive: capturing reasoning and language modeling, as well as LLMs and VLMs, there is also robustness study using corrupted prompts
- Theoretical results are compelling. The lemma results strengthen the paper's claim; not just a nice-to-have theory.
- The experimental results is honest about failure cases

Weaknesses:
- Intro claim mismatch: Research Question box in the intro: "Is it possible to perform routing in Sparse Mixture of
Experts (SMoEs) without a router?" it paints the picture that the method eliminates the need for routing entirely (a mindset I also had before actually delving into the method). But the method is a reweighting of router outputs, not entirely removing need of router
- Vision results are marginal. The improvements on CLIP-MoE (Table 3, Table 6) are mostly in the +0.2% to +1.1% range. These are small enough that statistical significance is questionable, yet no significance tests are reported for this section.
- The theoretical assumptions are strong. Lemma 3.1 assumes eigenvectors behave like random unit vectors on a high-dimensional sphere. This is a convenient assumption from random matrix theory, but trained expert weights are most probably not random; they've been optimized.

---

> ### Author Rebuttal · Authors · 2026-03-31
>
> Thank you for your time and your constructive feedback. We would like to address the concerns as follows:
>
> ```W1: Research Question box in the intro: "Is it possible to perform routing in Sparse Mixture of Experts (SMoEs) without a router?"```
>
> A1: We thank the reviewer for this observation. Our framework includes two settings:
>
> *   Router-augmented setting (SSMoE), where we reweight the original router using eigenvector-based signals.
> *   Router-free setting (EV), where routing is entirely replaced by eigenvector-based selection.
>
> The statement "routing without a router" refers specifically to the EV setting, which does not rely on a learned router. We will clarify this distinction in the introduction to avoid confusion.
>
>
> ```W2: Vision results are marginal```
>
> A3: We agree that absolute gains on clean vision benchmarks are modest due to the strong saturation of CLIP-style models. However, even +0.3-1.1 improvements are meaningful for a training-free method with no additional optimization cost.
>
> Importantly, the gains are consistent and statistically significant. On below Flickr30k results, paired bootstrap (n=1000) shows all improvements are positive with tight confidence intervals and p-values < 0.01, confirming reliability despite small magnitudes.
>
> | Data      | Metric  | Eval Method                              | Orig (SMoE) | SSMoE | GAP | 95% CI         | p-value |
> |-----------|---------|------------------------------------------|-------------|-------|-----|----------------|---------|
> | Flickr30k | I2T@1   | Paired Bootstrap (n=1000, seed=42)       | 61.8        | 62.3  | 0.5 | [0.15, 0.80]   | 0.001   |
> |           | I2T@5   |                                          | 83.4        | 84.1  | 0.6 | [0.40, 0.88]   | 0.000   |
> |           | I2T@10  |                                          | 89.6        | 89.9  | 0.3 | [0.06, 0.44]   | 0.008   |
>
>
> ```W3: Random Matrix Theory Assumption```
>
> A3: We thank the reviewer for this important point. The assumption is a standard analytical approximation for tractability, rather than a claim that trained weights are truly random. Prior work (Thamm et al., 2022; Staats et al., 2025) shows that while most eigenvectors follow random-like statistics, leading eigenvectors capture meaningful structure in neural representations. This supports using eigenvector-based signals as a principled basis for expert selection.
>
> Importantly, our method does not rely on this assumption strictly holding, empirical results across models and tasks demonstrate robustness to deviations from this idealized setting. We will clarify this and add supporting evidence in the revision.
>
> ```Q1: There is a pretty big drop on GSM8K GPT OSS 20B. Any intuition on failure cases?```
>
> A4: We observe that the drop on GSM8K is primarily caused by expert pruning (25%), which can remove task-critical experts (e.g., math-specialized experts). This is supported by the full-expert setting as below table, where SSMoE significantly outperforms both the baseline and the pruned variant (+6.6 vs Orig, +12.6 vs 75%). These results suggest that reasoning-heavy tasks are more sensitive to expert pruning, and that SSMoE itself remains effective when full expert capacity is preserved.
>
> | Task  | Orig | SSMoE(Full) | SSMoE(75% E) |
> |-------|------|-------------|--------------|
> | GSM8K | 36.8 | 43.4        | 30.8         |
>
> ```Q2: mentions a 5% GFLOPs reduction for EV Router vs. SMoE in Figure 4, does this include the one-time SVD cost or the overall inference latency comparison for SSMoE?```
>
> A5: The reported 5% GFLOPs reduction reflects inference-time computation only. The one-time SVD cost is not included, as it is a preprocessing step performed once and amortized over all subsequent inference. Therefore, it does not affect per-sample latency.

---

> > ### Author Rebuttal · Reviewer_T1GW · 2026-04-02
> >
> > thank you for the rebuttal. my concerns are mainly addressed and I am raising my score

---

> > > ### Author Response · Authors · 2026-04-02
> > >
> > > Dear Reviewer T1GW,
> > >
> > > We're glad our rebuttal addresses your concerns and appreciate that you increase your rating to 5.
> > >
> > > We will incorporate your suggestions into the revision of our paper as discussed. Please feel free to let us know if you have any further concerns.
> > >
> > > Best,
> > >
> > > The Authors

---

### Official Review · Reviewer_LpEm · 2026-03-10

**Soundness:** 3
**Presentation:** 2
**Significance:** 3
**Originality:** 3
**Overall Recommendation:** 5
**Confidence:** 3

**Summary:**

The authors propose routing experts based (partly) on the eigen-vectors of their weights and not only based on the learned router.

**Compliance With Llm Reviewing Policy:**

Affirmed.

**Final Justification:**

The rebuttal addressed my key concerns. The properties of EVs in MoE models this paper utilizes seem to me to be of interest to the ICML community. The evaluations are sound, given the additions over the rebuttal period. I recommend acceptance.

**Key Questions For Authors:**

Do you have some intuition on how it arises that the eigen vectors of MoEs have this property? Can you show that this will always happen? Under what assumptions?

Does the proposed expert pruning method result in good *perplexity* results?

What is the correlation between routing decisions of the proposed method (esp. with alpha=1.0) and of the learned router?

**Limitations:**

yes

**Strengths And Weaknesses:**

I find this work original, and potentially significant, however, the soundness and presentation are somewhat lacking. More detailed explanations below.

## Strengths
Suprising, useful and interesting property of eigenvectors of weight matrices in some MoE models.

Finding better MoE methods could be key to improving current SoTA LLMs and is an important topic to make progress on.

## Weaknesses
The organization of the paper is quite poor: The method (the key contribution of the paper) is abbreviated to such an extent that it cannot be understood without looking at the appendix (W_EV needs to be defined in the main paper)

Using accuracy to measure the quality of a lossy acceleration method (e.g. Fig.2) has been shown to be poor methodology ('Accuracy is not all you need' NeurIPS). Please use robust measures that cannot easily be manipulated, e.g., perplexity, KL-divergence or Flips.

To better understand why the proposed method works, it would be great to see the correlation between selected experts based on the router and on the eigen vectors.

Fig. 1 is poorly labelled (e.g. the location of OLMoE-7B and Phi-MoE labels are very odd)

The selection of models on which the method was evaluated is small and seems arbitrary.

The theoretical background is missing an explanation for how it comes to about the that the eigenvectors of experts can be used this way.

---

> ### Author Rebuttal · Authors · 2026-03-31
>
> We thank the Reviewer for the valuable feedback and would like to address concerns as below:
>
> ```W1 & W3: organization of the paper & Fig. 1 is poorly labelled```
>
> A1: We thank the reviewer for the feedback and agree that the presentation can be improved. We will revise the paper to provide a self-contained method description (including explicit definition of $W_{EC}$, reduce reliance on the appendix, and improve Figure 1.
>
> ```W2: measures that cannot manipulated, e.g., perplexity, KL-divergence or Flips.```
>
> A2: SSMoE yields more beneficial flips (W->C: 13.5%) than harmful ones (C->W: 9.2%), resulting in a net gain (+5.3%) as below table, showing improvements are driven by correcting errors rather than random changes.
>
> | Task       | Orig | SSMoE | GAP | FlipRate | C->W | W->C |
> |------------|------|-------|-----|----------|-----|-----|
> | ARC-C      | 44.2 | 52.1  | 7.9 | 15.7     | 7.3 | 8.5 |
> | ARC-E      | 77.9 | 82.4  | 4.5 | 8.7      | 3.4 | 5.3 |
> | BoolQ      | 62.8 | 72.4  | 9.6 | 39.2     | 12.6| 26.7|
> | GSM8K      | 36.8 | 30.8  | -6.0| 39.6     | 25.6| 14.0|
> | HellaSwag  | 29.0 | 37.5  | 8.5 | 12.1     | 2.4 | 9.7 |
> | OBQA       | 19.8 | 23.0  | 3.2 | 12.0     | 3.4 | 8.6 |
> | PIQA       | 62.5 | 72.2  | 9.7 | 21.6     | 5.1 | 16.4|
> | WinoGrande | 57.5 | 62.2  | 4.7 | 32.9     | 13.8| 19.1|
> | **Avg**    | **48.8** | **54.1** | **5.3** | **22.7** | **9.2** | **13.5** |
>
> ```W3 & Q3: correlation between selected experts based on the router and on the eigen vectors.```
>
> A3: SSMoE exhibits low overlap (0.1531) with the original router across all tasks, indicating that it selects substantially different experts rather than trivially mimicking the baseline. This suggests that the EV routing provides complementary selection signals, helping explain the observed performance gains.
>
> | Task       | Mean Overlap (EV vs Orig) |
> |------------|----------------------------------|
> | ARC-C      | 0.1581                           |
> | ARC-E      | 0.1586                           |
> | BoolQ      | 0.1526                           |
> | GSM8K      | 0.1581                           |
> | HellaSwag  | 0.1488                           |
> | OBQA       | 0.1462                           |
> | PIQA       | 0.1540                           |
> | WinoGrande | 0.1486                           |
> | **Mean**   | **0.1531**                       |
>
> ```W4: selection of models evaluated is small and seems arbitrary:```
>
> A4: Our model selection is neither small nor arbitrary. We evaluate SSMoE on state-of-the-art MoE-based LLMs across multiple scales (7B, 16B, 20B, 120B), covering both mid- and large-scale regimes. For vision, we include CLIP-MoE (~700M expert parameters) as a strong pretrained MoE baseline.
>
> This selection ensures coverage across model sizes, modalities (language and vision), and architectures, demonstrating the generality of our method.
>
> ```W5 & Q1: theoretical background is missing an explanation for how it comes```
>
> A5: Eigenvectors capture the principal directions of variation in expert weights, encoding dominant semantic features. By projecting routing signals onto these orthogonal directions, SSMoE provides a global, non-redundant basis for expert selection, reducing overlap and improving specialization (supported by Fig. 4).
>
> This intuition is grounded in prior work (Thamm et al., 2022), which shows that critical information in neural networks is concentrated in eigenvectors/eigenvalues. Our empirical analysis on MoE-based LLMs further confirms that eigenvectors encode rich semantic structure, motivating their use for routing in SSMoE.
>
>
> ```Q2: expert pruning method result in good perplexity results```
>
> A6: We agree that perplexity is an important metric for evaluating distributional quality. While our current evaluation focuses on downstream accuracy, SSMoE does not modify model weights and only changes routing, which preserves the underlying token distribution structure. Moreover, our flip analysis shows that improvements are dominated by W->C transitions, indicating better alignment rather than degradation. We will include perplexity results in the revised version.

---

> > ### Author Rebuttal · Reviewer_LpEm · 2026-04-02
> >
> > Thank you for your comments. Especially the overlap results are interesting.
> >
> > > Does the proposed expert pruning method result in good perplexity results?
> > >>  We agree that perplexity is an important metric for evaluating distributional quality.
> >
> > Currently, this key piece of information is missing: **Does the proposed method cause a serious output distribution shift?** Not knowing this **undermines the paper's soundness**. Please show the perplexity of your method and the perplexity of the unmodified network for a standard dataset like wikitext2 or c4 (if at all possible within the tight time frame please also add a baseline method for comparison).
> >
> > Relatedly, the flip rate, without comparison to other methods, is **unhelpful for understanding output distribution shift**, again raising questions about the **paper's soundness**.

---

> > > ### Author Response · Authors · 2026-04-03
> > >
> > > Dear Reviewer LpEm,
> > >
> > > Thank you for your thoughtful feedback and for acknowledging our previous responses. We agree that perplexity is a direct and appropriate metric to assess distribution shift.
> > >
> > > We report perplexity on WikiText-2 using GPT-OSS-20B:
> > >
> > > | Data      | Model        | Method   | PPL ↓ |
> > > |-----------|-------------|----------|--------|
> > > | Wikitext2 | GPT-OSS-20B | Original | 82.12  |
> > > |           |             | Router   | 75.91  |
> > > |           |             | SSMoE    | 73.23  |
> > >
> > > *We note that absolute PPL values are higher than standard LM benchmarks since GPT-OSS-20B is an instruction LLMs model, therefore, we focus on relative comparison under the same setup.*
> > >
> > > SSMoE consistently reduces perplexity compared to both the original model and the Router baseline, indicating improved alignment with data distribution rather than harmful distribution shift. The improvement over the Router baseline further shows that the gain is not solely due to routing modification, but from more effective expert selection.
> > >
> > > We will include these results in the revised version to address the reviewer's concern.

---

### Official Review · Reviewer_Jp5r · 2026-03-12

**Soundness:** 3
**Presentation:** 3
**Significance:** 2
**Originality:** 2
**Overall Recommendation:** 4
**Confidence:** 3

**Summary:**

This paper investigates the expert collapse problem in Sparse Mixture of Experts (SMoE) models and proposes SSMoE, a training-free framework that uses the eigenvectors of expert weight matrices as an alternative routing signal. The key observation is that these eigenvectors encode rich semantic information and are approximately orthogonal in high dimensions, which helps mitigate the representation collapse problem. The method is evaluated on reasoning benchmarks (GPT-OSS models), language embedding tasks (MTEB), and vision-language tasks (CLIP-MoE).

**Compliance With Llm Reviewing Policy:**

Affirmed.

**Key Questions For Authors:**

1. What is the wall-clock overhead of computing the SVD for all expert weight matrices, particularly for GPT-OSS-120B with 128 experts?
2. How is $α$ selected in practice? Is it tuned on a held-out validation set for each task? If so, how many evaluations does this require?
3. What happens when you apply SSMoE routing to the full model (all experts, no dropping) for the GPT-OSS experiments?

**Limitations:**

yes

**Strengths And Weaknesses:**

**Strengths**
1. The paper asks whether routing in SMoE can be performed without a learned router, which is a thought-provoking question. The empirical survey of collapse across ten models of varying scales (Figure 1) provides solid motivation.
2. The method requires no additional training or fine-tuning, which makes it practically appealing. The evaluation spans reasoning models, language models, and vision-language models, demonstrating breadth.
3. The headline result of outperforming the original GPT-OSS models while dropping 25% of experts and reducing memory by ~23% is noteworthy and practically significant.
4. Lemmas 3.1 and 3.2 provide clean and intuitive explanations for why eigenvector-based routing mitigates collapse. The near-orthogonality argument is simple but effective, and the empirical correlation plots (Figure 6) support it well.

**Weaknesses**
1. The GPT-OSS experiments simultaneously perform expert pruning (dropping 25% of experts) and change the routing strategy. The baselines (RandDrop, Router) also use 75% of experts. This means we cannot isolate the effect of SSMoE routing from the expert selection/dropping strategy. A cleaner experiment would compare SSMoE routing against the original router without any expert dropping, using all experts. The "6% improvement over original" claim is confounded by the fact that dropping poorly-chosen experts may itself be beneficial regardless of the routing mechanism.
2. The improvements on CLIP-MoE (Table 3) are quite small. It ranges +0.3% to +1.1% on clean data. While consistent, these are within the range of noise for many benchmarks. The zero-shot classification improvements (Table 6) are similarly marginal (+0.1% to +1.1%). This weakens the generalization claim.
3. The paper does not compare against other training-free routing strategies, such as hash-based routing or random routing with load balancing. For the expert dropping experiments, comparison with established MoE pruning methods beyond simple random dropping would strengthen the evaluation. The "Router" baseline (dropping by router similarity) is the only structured baseline, which is insufficient.

---

> ### Author Rebuttal · Authors · 2026-03-31
>
> We sincerely thank the reviewer for the constructive feedback and would like to address your concerns as follows:
>
> ``` W1 & Q3:SSMoE to the full model```
>
> A1: We thank the reviewer for the suggestion. We agree that isolating the routing effect from expert dropping is necessary for a fair evaluation. To address this, we conducted additional experiments on GPT-OSS-20B (10-shot) using all experts (no pruning). The results are reported below.
>
> SSMoE (full experts) improves over the original model by +6.4% on average, confirming that gains stem from the routing mechanism rather than expert pruning. It also slightly outperforms SSMoE (75% experts) by +1.1%, indicating that expert dropping is not the primary driver of improvement. In contrast, naive dropping (e.g., RandomDrop) degrades performance, further supporting that improvements arise from better expert utilization via SSMoE routing. These results demonstrate that SSMoE's gains are not driven by expert pruning, but by the proposed SSMoE mechanism.
>
> | Task       | Orig | SSMoE(Full) | SSMoE(75% E) |
> |------------|------|------|-----|
> | ARC-C      | 44.2 | 50.2 | 52.1 |
> | ARC-E      | 77.9 | 79.3 | 82.4 |
> | BoolQ      | 62.8 | 71.7 | 72.4 |
> | GSM8K      | 36.8 | 43.4 | 30.8 |
> | HellaSwag  | 29.0 | 40.9 | 37.5 |
> | OBQA       | 19.8 | 33.8 | 23.0 |
> | PIQA       | 62.5 | 63.9 | 72.2 |
> | WinoGrande | 57.5 | 58.3 | 62.2 |
> | **Avg**    | **48.8** | **55.2** | **54.1** |
>
> ```W2:improvements on CLIP-MoE are small```
>
> We agree that the absolute gains on clean vision retrieval benchmarks are modest. However, this is expected given the strength and maturity of CLIP-style models, where performance is already highly saturated. In such regimes, even improvements of 0.3-1.1 points are non-trivial, particularly for a training-free method that introduces no additional optimization cost.
>
> Importantly, the improvements are highly consistent: SSMoE outperforms CLIP-MoE across all reported metrics, datasets, and retrieval directions. Moreover, the gains become larger under corruption, suggesting that the primary benefit of our method is improved robustness rather than only boosting already-saturated clean performance.
>
> To further validate significance, we conduct paired bootstrap resampling (1000 samples). As shown below, all gains are positive with narrow confidence intervals and statistically significant p-values (< 0.025). These results indicate that the improvements are consistent and statistically reliable despite their small absolute magnitude.
>
> | Data | Metric | Orig | Ours | Δ | 95% CI       | p |
> |------|--------|------|------|---|--------------|---|
> | COCO | R@1    | 65.0 | 65.9 | +0.8 | [0.20,1.58] | .002 |
> |      | R@5    | 86.0 | 86.5 | +0.5 | [0.00,1.00] | .024 |
> |      | R@10   | 92.0 | 92.2 | +0.2 | [0.05,0.40] | .018 |
>
> ```W4:comparing hash-based routing```
>
> A4: SSMoE consistently outperforms both hash-based and random routing across all tasks, with a substantial average gain (+18.0% vs hash, +9.1% vs random).
>
> | Task       | Hash Router | Random Router | SSMoE |
> |------------|-------------|---------------|-------|
> | ARC-C      | 23.3        | 33.5          | 52.1  |
> | ARC-E      | 36.7        | 65.8          | 82.4  |
> | BoolQ      | 59.3        | 66.5          | 72.4  |
> | GSM8K      | 14.4        | 33.1          | 30.8  |
> | HellaSwag  | 29.2        | 29.7          | 37.5  |
> | OBQA       | 26.2        | 20.6          | 23.0  |
> | PIQA       | 53.2        | 57.2          | 72.2  |
> | WinoGrande | 46.9        | 53.3          | 62.2  |
> | **Avg**    | **36.1**    | **45.0**      | **54.1** |
>
> ```Q1:wall-clock overhead of computing SVD```
>
> A5:  SSMoE requires a one-time SVD computation during preprocessing; thus, it introduces no additional overhead at inference time (amortized O(1) per input).
> In practice, the SVD cost is modest even at scale: for GPT-OSS-120B, it takes 39.6s (approximate) or 3765s (deterministic) on two H200 GPUs. This one-time cost is negligible compared to model training and is amortized over all subsequent inference, making SSMoE practical for large-scale MoE models.
>
> ```Q2: how to select α ```
>
> A6: $\alpha$ controls the trade-off between the learned router and the orthogonal router. In practice, the method is robust across a wide range, with $\alpha \in [0.5, 0.9]$ consistently yielding strong performance. We tune $\alpha$ on a small validation set (e.g., 5-shot), and observe low sensitivity within this range.

---

> > ### Author Rebuttal · Reviewer_Jp5r · 2026-04-07
> >
> > Thank you for the rebuttal. I am maintaining my positive overall rating score.

---

### Decision · Program_Chairs · 2026-04-30

**Decision:**

Accept (spotlight)

**Comment:**

The paper show that the eigenvector of expert matrices can be used as a routing signal, avoiding the long-standing issue of expert collapse in MoEs.

The observation is interesting, and well supported by empirical evidence: eigenvector of pre-trained MoE can be used as routers.

The appreciation of experimental part is mixed: some reviewers found some setups unconvincing due to cofounding factors and scenarios that don't demonstrate a significant advantage over competitors, while other praised their extensiveness in different scenarios (language, VLM) and their strength. In particular, they are conducted are fairly large scale on pre-trained MoE, and show that the training-free approach work out of the box.

The simplicity of the approach, and its relevance, well supported by experiments and theoretical results, makes it a valuable contribution for the ICML community.

Some concerns were raised about readability / clarity / organisation of the paper. I recommend that authors integrate that feedback in the camera-ready version.